# LLM-Driven Treatment Effect Estimation Under Inference Time Text Confounding

**Yuchen Ma, Dennis Frauen, Jonas Schweisthal & Stefan Feuerriegel**
Munich Center for Machine Learning
LMU Munich
yuchen.ma@lmu.de

## Abstract

Estimating treatment effects is crucial for personalized decision-making in medicine, but this task faces unique challenges in clinical practice. At training time, models for estimating treatment effects are typically trained on well-structured medical datasets that contain detailed patient information. However, at inference time, predictions are often made using textual descriptions (e.g., descriptions with self-reported symptoms), which are incomplete representations of the original patient information. In this work, we make three contributions. (1) We show that the discrepancy between the data available during training time and inference time can lead to biased estimates of treatment effects. We formalize this issue as an *inference time text confounding* problem, where confounders are fully observed during training time but only partially available through text at inference time. (2) To address this problem, we propose a novel framework for estimating treatment effects that explicitly accounts for inference time text confounding. Our framework leverages large language models (LLMs) together with a custom doubly robust learner to mitigate biases caused by the inference time text confounding. (3) Through a series of experiments, we demonstrate the effectiveness of our framework in real-world applications.

## 1 Introduction

Estimating the *conditional average treatment effect (CATE)* is crucial for personalized medicine [14]. A growing number of machine learning methods have been developed for this purpose [e.g., 4, 6, 7, 28, 29, 33, 39, 42, 43, 47, 54, 62]. When estimating CATEs from observational data, it is necessary to account for confounders – i.e., variables that influence both treatment and outcome. If confounders are not properly adjusted for, the estimated treatment effects may *not* reflect the true causal effect and are thus *biased* [48].

The standard setting for CATE estimation typically assumes that the *same* set of confounders is observed at training time and at inference time [e.g., 39, 54]. However, this assumption is often violated in real-world clinical practice. At **training time**, comprehensive information about confounders is typically available via well-curated medical datasets, such as clinical registries or trial data, where structured patient records ensure that confounders are systematically recorded. At **inference time**, however, information about confounders may be only partially observed or entirely missing due to differences in data collection methods, resource constraints, or changes in clinical workflow. In other words, there is a discrepancy: at inference time, the CATE is often predicted from incomplete representations of the original patient information such as textual descriptions with self-reported symptoms.

39th Conference on Neural Information Processing Systems (NeurIPS 2025).

**Example (see Figure 1):** Consider a health insurance provider developing a medical chatbot to assist with treatment recommendations [55, 68]. The chatbot is trained on high-quality clinical datasets containing detailed patient histories, laboratory results, and physician assessments to ensure reliable predictions of treatment effects. However, at inference time, the chatbot relies primarily on free-text inputs from patients describing their symptoms, often without supporting diagnostic tests or clinical measurements. A similar challenge arises in emergency settings, such as during triage calls or ambulance dispatches, where only verbal symptom descriptions or brief text messages are available.

The above introduces a *discrepancy* between training time and test time: *confounders that were fully observed at training time are only partially available through unstructured text at inference time*, which can lead to biased treatment effect

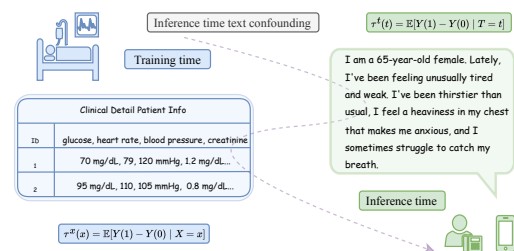

Figure 1: **Discrepancy between training time and inference time.** At training time, models for estimating treatment effects are typically trained on well-structured medical datasets that contain detailed patient information. However, at inference time (e.g., in telemedicine, remote healthcare consultations, or medical chatbots), predictions are often made using textual descriptions with self-reported symptoms. We formalize this discrepancy as *inference time text confounding*, where confounders are fully observed during training time but only partially available through text at inference time.

estimates. We later formalize this discrepancy as **inference time text confounding**. In other words, if inference time text confounding is *not* properly addressed, any treatment effects estimated in the above medical example would be *biased* and could potentially lead to harmful treatment decisions. To address this, we further propose a novel framework to compute unbiased estimates of the CATE, *even* in the presence of inference time text confounding.

Interestingly, the above setting involving inference time text confounding has not yet been studied. So far, standard methods for CATE estimation [e.g., 39, 54] require *identical* sets with confounders at both training time and inference and are therefore *not* applicable. Further, these methods focus on confounders with structured data, not text data. Other works focus on settings with confounding due to text data [e.g., 2, 10, 50, 61], yet these works again assume that confounders with text data are available during training and are therefore also *not* applicable.

To fill the above gap, we propose a novel framework for **unbiased CATE estimation in the presence of inference time text confounding**. We call our framework TCA (short for **t**ext **c**onfounding **a**djustment). Thereby, we aim to bridge the gap between advanced treatment effect estimation methods for text data and real-world medical constraints. Our TCA framework operates in three stages: ① estimate nuisance functions and construct pseudo-outcome based on the true confounders; ② generate surrogates of the text confounders by leveraging state-of-the-art large language models (LLMs); and ③ perform a doubly robust text-conditioned regression to obtain *valid* and *unbiased* CATE estimates. Crucially, we use LLMs *not* as causal reasoners – given their documented limitations in reliable inference [26] – but as a semantically rich text generator.

Intuitively, our TCA framework decouples treatment effect estimation (using true confounders) from inference time adjustments (using induced text confounders), which allows us to overcome the non-identifiability of CATE from text data alone. By conditioning on the generated text during training time, our framework learns to map the induced text confounders onto treatment effects, so that we can successfully remove the bias from having partially observed confounders at inference time. Further, our TCA is doubly robust and thus has favorable theoretical properties such as being robust to misspecification in the nuisance functions.

Overall, our **main contributions** are the following: [1] (1) We formalize the problem of inference time text confounding, which is common in real-world medical applications such as telemedicine, where full information about confounders is missing at inference time. (2) We develop a novel framework called TCA to adjust for inference time text confounding and provide unbiased treatment effect

---
[1]Code is available at `https://github.com/yccm/llm-tca`

estimations. (3) We demonstrate the effectiveness and real-world applicability of our framework across extensive experiments.

## 2 Related work

In the following, we focus on key literature streams relevant to our paper: (i) CATE estimation, (ii) treatment effect estimation from text, (iii) NLP for clinical decision support, and (iv) LLMs for causal inference. We provide an extended related work in Appendix A.

**CATE estimation:** Numerous methods have been proposed for estimating the CATE [e.g., 28, 29, 33, 39, 47, 54, 62]. On the one hand, there are works that suggest specific model architectures to address the covariate shift between treated and control groups by learning balanced representations [28, 54]. However, such specialized model architectures are not designed to handle unstructured data such as text. On the one hand, *meta-learners* offer a more flexible approach for constructing CATE estimators that can incorporate arbitrary machine learning models (e.g., neural networks) [7, 39]. Common meta-learners include the S-learner [39], T-learner [39], and the DR-learner [33]. Nevertheless, it is unclear how best to adapt meta-learners for complex settings such as those involving text data.

Of note, all of the above works operate in the *standard* CATE setting, where an *identical* set of confounders is observed at both training and inference time. However, if the confounding variables are different (e.g., if some confounders are only partially observed or hidden at inference time), standard methods will yield *biased* estimates that no longer reflect the true treatment effect [48]. Motivated by this issue, we focus on a specific setting that is characterized by such discrepancy where confounders were fully observed at training time but are only partially available through unstructured text data at inference time.

**Treatment effect estimation with text data:** A growing body of research has investigated causal inference with text data [e.g., 2, 8, 10, 13, 17, 31, 32, 50, 44, 45, 61, 65, 72, 70]). Therein, text data can have various roles: (a) text data can be *treatment variable* [e.g., 13, 44, 49, 58, 64, 66]; (b) text data can be a *mediator* [e.g., 17, 32, 61, 72]); and (c) text data can be an *outcome variable* [e.g., 16, 38, 57]). In contrast to that, our setting considers text data as a *confounder*.

Several works have also explored (d) text data as a *confounder* [e.g., 2, 8, 31, 45, 50, 65, 70]). To adjust for text confounding, recent methods typically attempt to mitigate the underlying confounding bias by viewing text features as proxies for unobserved confounders [2, 10, 50, 61].

However, these works are *orthogonal* to our setting. In the standard proxy-based literature, the confounding variable is unobserved, but proxies are assumed to be *available at both training and inference*. In contrast, our setting introduces a unique challenge due to inference time text confounding: the proxy (= text data) is *unavailable during training* but are only available at inference time. Hence, proxy-based approaches are not applicable to our setting. See the detailed comparison in Appendix D.

**NLP for clinical decision support:** Natural language processing (NLP) has become a useful tool for clinical decision support, such as by enabling to extract information from unstructured clinical texts [e.g., 3, 9, 25, 73, 67, 52, 21, 23, 24]). Early work focused on using NLP to process clinical notes and symptom descriptions for some basic analysis and classification tasks, while more recent approaches leverage deep learning for learning the representations, particularly transformer-based models like BERT and GPT [1, 11, 12, 25, 40]. In a similar vein, LLMs have shown promise in medical applications, such as clinical text summarization, diagnosis, and treatment recommendations [12, 46, 56, 59, 60].

However, most NLP models – including LLMs – are designed for *associative* learning rather than *causal* reasoning [26, 27]. Hence, LLM fails in addressing text confounding and would thus give *biased* estimates. This is unlike our TCA framework where we aim to give *unbiased* estimates.

**LLMs for causal inference:** Recent works have explored the capabilities of LLMs in causal reasoning across various causal tasks [e.g., 18, 26, 27, 37, 69, 71]. For example, [12] leverages LLMs to impute missing data for causal inference. However, recent works by [26, 27] demonstrate that LLMs fail to reliably reason about causality, which raises concerns given the need for reliable inferences, particularly in medicine [35], where flawed causal reasoning of LLM can lead to harmful or even dangerous decisions.

In sum, the above works are *orthogonal* to our work. We are not aiming to use LLMs for causal reasoning, given their limitations in reliable inference. Instead, we leverage LLMs as an *auxiliary tool*: a semantically rich text generator in an intermediate step of our framework.

**Differences to other settings:** The main difference between our setting and other settings lies in the availability and role of confounders during training and inference time, see Table 1.

(i) *Comparison with the standard CATE estimation setting:* Unlike the standard CATE settings [e.g., 39, 54], where the same set of confounders is observed at both training and inference, our setting involves a true confounder $X$ that is observed during training but *not* inference (while some induced text confounder $T$ is *un*observed during training but observed at

Table 1: Comparison of confounder availability across different settings. $X$: true confounder; $T$: induced text confounder.

| Setting | Training time | | Inference time | |
|---|---|---|---|---|
| | $X$ | $T$ | $X$ | $T$ |
| Standard CATE setting | ✓ | ✗ | ✓ | ✗ |
| Proxy-based setting | ✗ | ✓ | ✗ | ✓ |
| Our novel setting | ✓ | ✗ | ✗ | ✓ |

✗: Not available; ✓: Available.

inference). (ii) *Comparison with the proxy-based setting:* In the proxy-based setting [2, 10, 50, 61], one views text $T$ as a noisy proxy of true confounder $X$, which implies that the true confounders $X$ should be assumed latent and are *never* directly observed, while the proxy $T$ should *always* be observed. By contrast, in our setting, $T$ is *not* a proxy but an induced variable from $X$, because $T$ encodes partial information of the confounders. $T$ is *unobserved* during training but *fully observed* during inference, whereas $X$ is *observed* during training but *unobserved* during inference, which is **opposite** to the proxy setting.

**Research gap:** We focus on CATE estimation for clinical decision support where confounders are only partially available through text at inference time. To the best of our knowledge, we are the first to formalize the underlying issue as inference time text confounding and propose a novel framework tailored to give unbiased CATE estimation.

# 3 Problem setup

**Setting:** We consider a treatment $A \in \mathcal{A} = \{0, 1\}$ (e.g., an anti-hypertension drug) and the outcome of interest $Y \in \mathcal{Y} \subseteq \mathbb{R}$ (e.g., cardiovascular events), for which we want to estimate the treatment effect. In our setting, we further consider the following discrepancy between training time and inference time:

**Training time:** Here, we observe confounders $X \in \mathcal{X} \subseteq \mathbb{R}^{d_x}$. For example, this could be well-curated information from electronic health records such as blood pressure readings, cholesterol levels, and kidney function tests. During training, we thus have access to an observational dataset $\mathcal{D}_X = (x_i, a_i, y_i)_{i=1}^n$ sampled i.i.d. from the unknown joint distribution $\mathbb{P}(X, T, A, Y)$.

**Inference time:** Here, we do not observe the confounders $X$. Rather, we observe some textual representation (e.g., a self-description of symptoms such as "I feel dizzy" or "I have headaches") but without access to laboratory results or prior prescriptions. Formally, we observe some induced text confounders $T$, while the true confounders $X$ are unavailable. The induced text confounders $T \in \mathcal{T} \subseteq \mathbb{R}^{d_t}$ are generated by $X$, which are *unobserved* during training but *observed* at inference time. The test dataset is $\mathcal{D}_T = (t_i, a_i, y_i)_{i=1}^n$ and also sampled i.i.d. from the same joint distribution $\mathbb{P}(X, T, A, Y)$.

The underlying causal graph for our setting is shown in Fig. 2. We refer to the above discrepancy as *inference time text confounding*.

**Clinical relevance:** Our setting is highly relevant in medical practice. One application is telemedicine [19] where CATE models are trained on well-curated medical datasets (e.g., trial data) containing comprehensive confounder information (where $X$ includes patient characteristics and detailed diagnostic features, different risk factors, e.g., age, gender, prior diseases). Hence, at training time, detailed information about confounders is available. In contrast, at inference time, detailed patient information is unavailable, simply because diagnostic facilities are not available in telemedicine. Hence, at inference time, the CATE model can only be applied to symptom descriptions from patients (where $T$ provides partial information about the patient characteristics and diagnostic features in $X$).

**Notation:** We use capital letters to denote random variables and small letters for their realizations from corresponding spaces. We denote the propensity score by $\pi_a^x(x) = \Pr(A = a \mid X = x)$, which is the treatment assignment mechanism in the observational data. We denote the response surfaces via $\mu_a(x) = \mathbb{E}[Y \mid X = x, A = a]$. Importantly, we later use superscripts to distinguish between training/inference: we use $\pi_a^x$, $\mu_a^x$ to refer to nuisance functions related to the true confounder $X$ and $\pi_a^t$, $\mu_a^t$ for nuisance functions related to the induced text confounder $T$.

**Potential outcomes framework:** We build upon Neyman-Rubin potential outcomes framework [51]. Hence, $Y(a) \in \mathcal{Y}$ denotes the potential outcome for a treatment intervention $A = a$. We have two potential outcomes for each individual: $Y(1)$ if the treated (i.e., $A = 1$), and $Y(0)$ if not treated (i.e., $A = 0$). However, due to the fundamental problem of causal inference [22], only one of the potential outcomes is observed. Hence, $Y = AY(1) + (1 - A)Y(0)$. The *conditional average treatment effect* (CATE) with respect to confounder $X$ is defined as $\tau(x) = \mathbb{E}[Y(1) - Y(0) \mid X = x]$, which is the expected treatment effect for an individual with covariate value $X = x$.

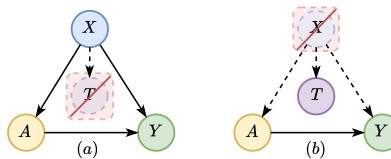

Figure 2: **Causal graph for inference time text confounding.** (a) At *training time*, we have access to the true confounders $X$ but the induced test confounders $T$ are unobserved. (b) At *inference time*, the induced text confounders $T$ are observed, while the true confounders $X$ are unavailable.

We follow previous literature [e.g., 6, 33, 54, 62] to make the following standard assumptions to ensure the identifiablity of the CATE from observational data.

**Assumption 3.1.** (i) Consistency: if $A = a$, then $Y = Y(a)$; (ii) Unconfoundedness: $Y(0), Y(1) \perp A \mid X$; (iii) Overlap: $\Pr(0 < \pi_a^x(x) < 1) = 1$.

Of note, the above assumptions are standard in causality [48]. Consistency ensures that the observed outcome $Y$ aligns with the potential outcome $Y(a)$ under the assigned treatment $A = a$. Unconfoundedness implies there are no unobserved confounders, i.e., all factors influencing both the treatment and the outcome are captured in the observed covariates $X$. Overlap guarantees that the treatment assignment is non-deterministic, i.e., that every individual has a non-zero probability of receiving each treatment option.

In practice, it is natural to view the text as a noisy 'measurement' of the true confounder. We thus formalize the induced text confounder $T$ generated from $X$.

**Assumption 3.2** (Text generation mechanism). There exists a measurable function $h : \mathcal{X} \times \mathcal{E} \to \mathcal{T}$, and a random variable $\epsilon$ with support $\mathcal{E}$, such that the induced text confounder $T$ is generated from the true confounder $X \in \mathcal{X}$ as $T = h(X, \epsilon)$, where the noise variable satisfies $\epsilon \perp\!\!\!\perp Y(a) \mid X$, i.e., the noise is independent of the potential outcomes conditional on the true confounder $X$.

**Objective:** For ease of readability, we use different colors to distinguish the CATE with respect to true confounder $X$, $\tau^x(x)$, and the CATE with respect to induced text confounder $T$, $\tau^t(t)$. In the standard setting, $\tau^x(x)$ is often of interest. However, here, our target is $\tau^t(t)$. At inference time, we focus on estimating $\tau^t(t) = \mathbb{E}[Y(1) - Y(0) \mid T = t]$, which is the expected treatment effect for an individual with covariate value $T = t$.

# 4 Our novel TCA framework to adjust for inference time text confounding

**Overview:** In this section, we introduce our **t**ext **c**onfounding **a**djustment (TCA) framework for addressing inference time text confounding. We first discuss the challenges of CATE estimation in inference time text confounding (Sec. 4.1), where we demonstrate the limitation of naïve methods due to estimation bias and thereby motivate the need for a novel method to achieve unbiased estimation. We then present our text generation (Sec. 4.2) and doubly-robust CATE estimation procedure (Sec. 4.3). Our full TCA framework is given in Sec. 4.4.

## 4.1 Adjusting for inference time text confounding

*Why is CATE estimation non-trivial in our setting?* Estimating the CATE in inference time text confounding is very challenging. Recall that the CATE with respect to true confounder $X$, $\tau^x(x)$,

and the CATE with respect to induced text confounder $T$, $\tau^t(t)$, are different. At training time, under Assumption 3.1, the CATE $\tau^x(x)$ can be directly identified from observational data $\mathcal{D}_X = (x_i, a_i, y_i)_{i=1}^n$ as $\tau^x(x) = \mu_1^x(x) - \mu_0^x(x)$. However, *the target estimand* at inference time, $\tau^t(t)$, is *non-identifiable* from observational test data $\mathcal{D}_T = (t_j, a_j, y_j)_{j=1}^m$ through $T$ alone.

*How do naïve baselines work?* A naïve baseline would attempt to directly estimate $\tau^t(t)$ from $\mathcal{D}_T$. Assuming that $T$ would be observed during training, a model is trained to learn the response surface with respect to $\mu_a^t(t)$ that gives $\mathbb{E}[Y \mid A = a, T = t]$. Here, a naïve baseline directly estimates $\tau_{\text{naïve}}^t(t) = \mathbb{E}[Y \mid A = 1, T = t] - \mathbb{E}[Y \mid A = 0, T = t]$.

*Why do naïve baselines fail?* A naïve method estimates $\tau_{\text{naïve}}^t(t)$ which is not equal to the true CATE $\tau^t(t)$. The induced text confounder $T$ generated from $X$ often does not capture all the necessary confounding information in $X$ for potential outcomes. As a result, there exists confounding information in the *residual confounders* $X \backslash T$ that affects both $A$ and $Y$, which are unobserved in $\mathcal{D}_T$. Due to the residual confounding, we have $\mathbb{E}[Y \mid A = a, T = t] \neq \mathbb{E}[Y(a) \mid T = t]$. Below, we state the pointwise confounding bias of the naïve baseline following [5].

**Lemma 4.1** (Pointwise confounding bias of the naïve baseline)**.** *For any $t \in \mathcal{T}$, under Assumption 3.1, the naïve baseline estimating $\tau_{\text{naïve}}^t(t)$ has pointwise confounding bias with respect to the true CATE $\tau^t(t)$, given by*

$$
\begin{aligned}
\text{bias}(t) &= \tau_{\text{naïve}}^t(t) - \tau^t(t) \\
&= \Big( \mathbb{E}\big[\mu_1^x(X) \mid A = 1, T = t\big] - \mathbb{E}\big[\mu_1^x(X) \mid T = t\big] \Big) \\
&\quad - \Big( \mathbb{E}\big[\mu_0^x(X) \mid A = 0, T = t\big] - \mathbb{E}\big[\mu_0^x(X) \mid T = t\big] \Big).
\end{aligned}
\tag{1}
$$

*Proof.* The proof is in the Appendix B.1. $\qquad\square$

In the context of inference time text confounding, when $T$ does not capture all the necessary confounding information in $X$ for potential outcome, the naïve estimator is necessarily biased, as

$$
\mathbb{E}\big[\mu_a^x(X) \mid A = a, T = t\big] \neq \mathbb{E}\big[\mu_a^x(X) \mid T = t\big]
\tag{2}
$$

*Remark* 4.2 (Non-zero bias of the naïve estimator)**.** The bias of the naïve estimator remains non-zero under Assumption 3.1 and $Y(0), Y(1) \not\perp A \mid T$.

*How to properly adjust for inference time text confounding?* To address the challenges due to inference time text confounding, we reformulate the target estimand $\tau^t(t)$. Our solution leverages the fact that we have access to the true confounder $X$ in the training data to circumvent the problems arising from the test data with $T$ alone.

**Lemma 4.3** (Identifiablity of $\tau^t(t)$)**.** *For any $t \in \mathcal{T}$, under Assumption 3.2, $\tau^t(t)$ can be identified through $\tau^x(x)$ via*

$$
\tau^t(t) = \mathbb{E}\left[\tau^x(X) \mid T = t\right].
\tag{3}
$$

*Proof.* Proof is in the Appendix B.2. $\qquad\square$

Lemma 4.3 gives us *unbiased* estimation of $\tau^t(t)$ through $\tau^x(x)$. Given the identifiability of $\tau^x(x)$, we can estimate $\tau^t(t)$ via

$$
\tau^t(t) = \mathbb{E}\left[\mu_1^x(X) - \mu_0^x(X) \mid T = t\right]
\tag{4}
$$

Importantly, the above reformulation motivates our approach: the reformulation essentially addresses the estimation challenge by first learning the ground-truth response surfaces $\mu_a^x$ from $\mathcal{D}_X$ and then *conditioning* on the inference time text confounder $T$. Eq. 4 allows us to construct a tailored training procedure where we first fit response surfaces $\mu_a^x$ using $\mathcal{D}_X$, and then perform text-conditioned CATE regression by training an auxiliary model to estimate $\mathbb{E}[\mu_a^x(X) \mid T = t]$. In this way, at training time, the model learns the function mapping from text confounder $T$ to the true CATE $\tau^t(t)$. Once finishing training the model, at test time when $X$ is unobservable, the model still ensures unbiased CATE estimation in the presence of inference time text confounding.

## 4.2 Text-based surrogate confounder

The above analysis assumes access to the induced text confounder $T$ during training. However, in the practical setting, we only have access to the confounder $X$. Hence, to be able to nevertheless leverage our reformulation in Lemma 4.3, we construct a text-based surrogate confounder $\tilde{T} = g(X)$ through an LLM-based *text generation* procedure, where $g : \mathcal{X} \to \mathcal{T}$ maps structured features to text space. Importantly, by using $g$ in this way, we map confounder $X$ to the induced text confounder $\tilde{T}$ naturally follows the text generation mechanism in Assumption 3.2, as the only input to the LLM is the $X$, thus the key confounding information in $X$ is carried over into $\tilde{T}$ with some random noise from the LLM.

Note that our method does not require the induced text confounder $T$ to contain all the necessary information from the true confounder $X$ for potential outcomes to ensure unbiased estimation of $\tau^t(t)$. Instead, we allow $T$ to preserve only partial confounding information in $X$, which is essentially consistent with real-world scenarios, where information about true confounding at inference time can be partially observed or missing. For instance, if $X$ includes diagnostic measurements such as heart rate, $\tilde{T}$ might be a self-reported description of the symptoms such as *"My heart is racing"*.

We thus construct our training data $\tilde{\mathcal{D}}_X = \left(x_i, \tilde{t}_i, a_i, y_i\right)_{i=1}^n$. The generation details can be found in Sec. 5 and Appendix C. The surrogate $\tilde{T}$ enables us to adapt the key identity from Sec. 4.1, that $\tau^t(t)$ can be computed by $\mathbb{E}\left[\mu_1^x(X) - \mu_0^x(X) \mid \tilde{T} = \tilde{t}\right]$. This text generation step transforms our theoretical identifiability result into a practical framework [2]. In Sec. 6, we empirically validate that including this step helps estimate $\tau^t(t)$ more effectively than methods ignoring residual confounding.

## 4.3 Doubly-robust CATE estimation with text confounders

Here, we adapt a state-of-the-art CATE meta-learner for our framework. Specifically, we leverage a doubly-robust (DR) learner [33] for estimating CATE $\tau^t(t)$ due to the favorable theoretical property of being double robust. This step has two sub-steps: (i) Nuisance functions estimation using the true confounder $X$ and pseudo-outcomes construction; and (ii) text-conditioned regression with pseudo-outcomes.

**Estimating nuisance functions and pseudo-outcomes:** First, we estimate the response surface $\mu_a^x(x)$ and the propensity score $\pi^x(x)$ using training data $\mathcal{D}_X$. Let $\hat{\eta}^x(x) = (\hat{\mu}_0^x(x), \hat{\mu}_1^x(x), \hat{\pi}^x(x))$ denote the estimated nuisance functions, where $\hat{\mu}_a^x(x)$ is the estimated response surface for treatment $A = a$, $\hat{\pi}^x(x)$ is the estimated propensity score. Following [5], these estimates enable the construction of a doubly-robust pseudo-outcome for each observation sample via

$$\tilde{Y}_i = \hat{\mu}_1^x(x_i) - \hat{\mu}_0^x(x_i) + \frac{A_i}{\hat{\pi}^x(x_i)}(Y_i - \hat{\mu}_1^x(x_i)) - \frac{1 - A_i}{1 - \hat{\pi}^x(x_i)}(Y_i - \hat{\mu}_0^x(x_i)). \qquad (5)$$

The first term estimates the CATE conditioned on $X$, while the remaining terms apply inverse propensity weighting to correct for potential biases in the response surface estimates.

**Text-conditioned regression:** To estimate $\tau^t(t)$ using the auxiliary dataset $\tilde{\mathcal{D}}_X$ from Sec. 4.2, we map the text $\tilde{T}$ to the text embedding $\phi(\tilde{T})$ using a pretrained LLM. We then train a regression model $f_\theta : \mathcal{T} \to \mathbb{R}$ to predict the pseudo-outcome $\tilde{Y}$ from text embeddings $\phi(\tilde{T})$ by minimizing the loss

$$\mathcal{L}(\theta) = \sum_{i=1}^n \left(\tilde{Y}_i - f_\theta\left(\phi(\tilde{t}_i)\right)\right)^2 + \lambda\|\theta\|_2^2, \qquad (6)$$

where $\lambda$ is a regularization parameter. At inference time, the final CATE estimate is given by $\hat{\tau}^t(t) = f_\theta(\phi(t))$, which maps text inputs to CATE estimates.

**Corollary 4.4** (Double robustness property of the estimator). *The estimator satisfies the double robustness property by construction: $\hat{\tau}^t(t)$ converges to the true $\tau^t(t)$ if either (i) the response surface estimates $\hat{\mu}_a^x$ are consistent, or (ii) the propensity score estimate $\hat{\pi}^x$ is consistent. This ensures validity even under potential model misspecification.*

---

[2] Modern LLMs' ability to transfer structured data into semantically rich text provides a principled approximation. While discrepancies of distribution between $\tilde{T}$ and the hypothetical $T$ may exist, it can be mitigated by some domain adaptation methods. For example, if we have access to the example $T$, we can adapt $\tilde{T}$ accordingly.

We refer to the Appendix B.3 for the detailed proof.

### 4.4 TCA framework

**Training:** Our framework text confounding adjustment (TCA) operates through three stages. *Stage ①: Estimating nuisance functions and pseudo-outcome*. We first estimate the nuisance functions based on $\mathcal{D}_X$ and construct doubly-robust pseudo-outcome. *Stage ②: Generating the text-based surrogate confounder*. We generate text surrogates $\tilde{T}$ through an LLM-based mapping, which allows us to link the true confounder to the induced text confounder. *Stage ③: Doubly-robust text-conditioned regression*. We perform a doubly-robust CATE estimation, where we train the regression model with pseudo-outcome conditioned on text to estimate CATE. The full TCA is shown in Algorithm 1.

**Inference:** At inference time, we are given a new text sample $t$ from $\mathcal{D}_T$ and compute CATE via $\hat{\tau}^t(t) = f_\theta(\phi(t))$.

Therein, we address inference time text confounding by reformulating the target estimand with the true confounder $X$ via Eq. 4, which establishes the *theoretical foundation* for our method in CATE estimation: we decouple the treatment effect estimation (using $X$) from the test-time adaptation (using $T$), which allows us to overcome the non-identifiability of $\tau^t(t)$ from $\mathcal{D}_T$ alone. By conditioning on the generated $\tilde{T}$ during training, our framework allows us to learn a mapping of the induced text confounder onto the treatment effects.

## 5 Implementation details

Given structured clinical confounders $X$, we generate text confounders $\tilde{T}$ via the OpenAI API [1]. The generated narratives are approximately 150-200 tokens long. We obtain representations of $\tilde{T}$ using pretrained BERT [11]. Token embeddings from the final transformer layer undergo mean pooling, yielding fixed-dimensional representations $\phi(\tilde{t}_i) \in \mathbb{R}^{768}$. Training time and further implementation details are in Appendix C.

## 6 Experiment

### 6.1 Setup

**Datasets:** We use the following datasets from medical practice for benchmarking: (i) The International Stroke Trial (**IST**) [53] is one of the largest randomized controlled trials in acute stroke treatment. The dataset comprises $19,435$ patients. (ii) **MIMIC-III** [30] is a large, single-center database comprising information relating to patients admitted to critical care units at a large tertiary care hospital. MIMIC-III contains $38,597$ distinct adult patients. We adhere to the terms and conditions governing the use of the MIMIC dataset (in particular, our analysis is HIPAA compliant). Details are in Appendix E.

---

**Algorithm 1:** TCA for CATE estimation with inference time text confounding.

---

1   **Input:** $\mathcal{D}_X = \{(x_i, a_i, y_i)\}_{i=1}^n$, test data $\mathcal{D}_T = (t_j, a_j, y_j)_{j=1}^m$, text generator $g$, pretrained encoder $\phi$, regularization $\lambda$

2   **Training:**

3   ▷ *Stage ①*

4   Estimate nuisance functions on $\mathcal{D}_X$:

5     • Fit   $\hat{\mu}_a^x(x) \leftarrow$ $\arg\min_\mu \sum_{i=1}^n \left(y_i - \mu_a(x_i)\right)^2 \;\; \forall a \in \{0,1\}$

6     • Fit $\hat{\pi}^x(x) \leftarrow \arg\max_\pi \sum_{i=1}^n \Big[a_i \log \pi(x_i) + (1-a_i)\log\big(1-\pi(x_i)\big)\Big]$

7   **for** $i = 1$ **to** $n$ **do**

8     |   Construct doubly robust pseudo-outcomes

9     |   $\tilde{y}_i \leftarrow \hat{\mu}_1^x(x_i) - \hat{\mu}_0^x(x_i) + \frac{a_i}{\hat{\pi}^x(x_i)}\big(y_i - \hat{\mu}_1^x(x_i)\big) - \frac{1-a_i}{1-\hat{\pi}^x(x_i)}\big(y_i - \hat{\mu}_0^x(x_i)\big)$

10   ▷ *Stage ②*

11   **for** $i = 1$ **to** $n$ **do**

12     |   Generate text surrogates $\tilde{t}_i \leftarrow g(x_i)$

13     |   Get text embeddings $z_i \leftarrow \phi(\tilde{t}_i)$

14   ▷ *Stage ③*

15   Train CATE learner $f_\theta$ via $f_\theta \leftarrow \arg\min_\theta \left[ \sum_{i=1}^n \left(\tilde{y}_i - f_\theta(z_i)\right)^2 + \lambda\|\theta\|_2^2 \right]$

16   **Inference:**

17   Predict CATE given new text input $t_j \in D_T$ via $\hat{\tau}^t(t_j) \leftarrow f_\theta\big(\phi(t_j)\big)$

18   **Output:** CATE estimator $\hat{\tau}^t(t)$

---

Due to the fundamental problem of causal inference, the counterfactual outcomes are never observed in real-world data. We thus follow prior literature (e.g.,[6, 7, 33, 39, 54]) and benchmark our model using semi-synthetic datasets. Details of datasets are in Appendix D.

**Baselines:** Due to the novelty of our setting, there are no existing methods tailored for this inference time text confounding setting. Hence, we compare our method against the naïve **text-based estima-**

Figure 3: Performance of CATE estimation under varying confounder strengths and prompt strategies across datasets.

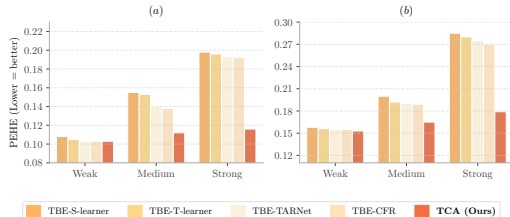

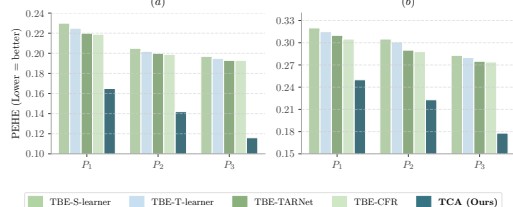

(a) Results for CATE estimation under varying confounder strengths. (a): IST dataset. (b): MIMIC-III dataset.

(b) Results for CATE estimation under varying prompt strategies. (a): IST dataset. (b): MIMIC-III dataset.

Table 2: Performance of CATE estimation evaluated by PEHE across different demographic subgroups for each dataset. Reported: mean $\pm$ standard deviation.

| | IST | | MIMIC-III | | | IST | | MIMIC-III | |
|---|---|---|---|---|---|---|---|---|---|
| | $G_M$ | $G_F$ | $G_M$ | $G_F$ | | $G_Y$ | $G_O$ | $G_Y$ | $G_O$ |
| TBE-S-learner [39] | 0.197 $\pm 0.03$ | 0.201 $\pm 0.04$ | 0.282 $\pm 0.02$ | 0.285 $\pm 0.03$ | TBE-S-learner [39] | 0.203 $\pm 0.04$ | 0.197 $\pm 0.03$ | 0.288 $\pm 0.03$ | 0.283 $\pm 0.02$ |
| TBE-T-learner [39] | 0.195 $\pm 0.03$ | 0.197 $\pm 0.04$ | 0.279 $\pm 0.02$ | 0.28 $\pm 0.03$ | TBE-T-learner [39] | 0.201 $\pm 0.03$ | 0.195 $\pm 0.03$ | 0.282 $\pm 0.03$ | 0.279 $\pm 0.02$ |
| TBE-TARNet [54] | 0.192 $\pm 0.03$ | 0.193 $\pm 0.04$ | 0.275 $\pm 0.02$ | 0.277 $\pm 0.03$ | TBE-TARNet [54] | 0.198 $\pm 0.03$ | 0.192 $\pm 0.03$ | 0.278 $\pm 0.03$ | 0.274 $\pm 0.02$ |
| TBE-CFR [54] | 0.191 $\pm 0.03$ | 0.193 $\pm 0.04$ | 0.274 $\pm 0.02$ | 0.275 $\pm 0.03$ | TBE-CFR [54] | 0.195 $\pm 0.03$ | 0.191 $\pm 0.03$ | 0.275 $\pm 0.03$ | 0.273 $\pm 0.02$ |
| **TCA** (Ours) | **0.115** $\pm 0.02$ | **0.116** $\pm 0.02$ | **0.179** $\pm 0.02$ | **0.180** $\pm 0.02$ | **TCA** (Ours) | **0.117** $\pm 0.03$ | **0.115** $\pm 0.02$ | **0.181** $\pm 0.03$ | **0.178** $\pm 0.02$ |

Lower = better (best in bold).     Lower = better (best in bold).

**tors (TBE)**: training standard CATE learners directly on the text features $\phi(\tilde{T})$. We compare with the TBE-T-learner [39]; TBE-S-learner [39]; TBE-TARNet [54]; TBE-CFRNet [54]. Note that we use the same $\tilde{T}$ and $\phi(\tilde{T})$ for both our method and the baselines to ensure a *fair comparison*. We also used equivalent hyper-parameters across all baselines, thus ensuring that baselines can have the same number of hidden layers and units. (See implementation details in Appendix C.)

**Evaluation metrics:** We report the *precision of estimating the heterogeneous effects (PEHE)* criterion [6, 20] to evaluate the performance of our framework in estimating the CATE. We reported results for 5 runs each.

## 6.2 Results

**Benchmarking results for CATE estimation:** We report the performance of CATE estimation in Table 3. Our TCA consistently outperforms the baseline methods across both datasets and groups. The improvement is substantial, as the error rates of our TCA are approximately 40% and 35% lower than the best baseline methods on IST and MIMIC-III, respectively. This demonstrates the effectiveness of our framework in mitigating bias from residual confounding at inference time.

**Additional studies:** We further conduct additional studies to gain a deeper understanding of our framework.

• **Varying confounder strengths:** We evaluate the performance of TCA in estimating CATE under different confounding strengths by varying the influence of true confounders $X$ to assess the impact of residual confounding on estimation accuracy. Results are in Fig. 3a. It demonstrates the effectiveness and robustness of our framework under varying confounder strength.

Table 3: Benchmarking results for CATE estimation evaluated using PEHE on the IST and MIMIC-III datasets. Reported: mean $\pm$ standard deviation.

| | IST | MIMIC-III |
|---|---|---|
| TBE-S-learner [39] | 0.198 $\pm 0.03$ | 0.283 $\pm 0.02$ |
| TBE-T-learner [39] | 0.196 $\pm 0.03$ | 0.280 $\pm 0.02$ |
| TBE-TARNet [54] | 0.193 $\pm 0.03$ | 0.275 $\pm 0.02$ |
| TBE-CFR [54] | 0.192 $\pm 0.03$ | 0.274 $\pm 0.02$ |
| **TCA** (Ours) | **0.116** $\pm 0.02$ | **0.179** $\pm 0.02$ |

Lower = better (best in bold).

• **Varying prompt strategies:** We further analyze the impact of prompt engineering on LLM-based generation by evaluating three different prompt strategies: (i) *Factual prompts* ($P_1$) are a basic approach using a fixed template. Here, the prompts directly convert the patient information using a fixed structure (e.g., "`Transfer this patient information into a paragraph of text.`") (ii) *Narrative prompts* ($P_2$) allow to capture rich context. Here, the prompts capture detailed symptom experiences but exclude diagnostics (e.g., "`Write a detailed clinical narrative for a patient with these features.`") (iii) *Symptom-focused prompt* ($P_3$) (advanced): Patient-centric symptom descriptions (e.g., "`Now this patient just does not have diagnostic measurements available. How would this patient`

`describe his/her feeling by text?`"). Detailed examples of these prompts are provided in Appendix D.4. Results are in Fig 3b. Evidently, we see our method outperform baselines across all prompt strategies, confirming the effectiveness of TCA.

• **Varying LLMs:** We further analyze the impact of different LLMs. We show the results for benchmarking CATE estimation with PEHE on IST and MIMIC-III datasets using GPT-3.5 Turbo in Table 4. It shows that our method still performs better than the baselines regardless of the LLMs used for generating text.

Table 4: Results of CATE estimation benchmarking evaluated with PEHE on the IST and MIMIC-III datasets under varying LLMs. Reported: mean $\pm$ standard deviation.

|  | IST | MIMIC-III |
|---|---|---|
| TBE-S-learner [39] | 0.224 $\pm 0.04$ | 0.314 $\pm 0.03$ |
| TBE-T-learner [39] | 0.221 $\pm 0.04$ | 0.310 $\pm 0.03$ |
| TBE-TARNet [54] | 0.218 $\pm 0.04$ | 0.304 $\pm 0.03$ |
| TBE-CFR [54] | 0.216 $\pm 0.04$ | 0.299 $\pm 0.03$ |
| **TCA** (Ours) | **0.141** $\pm 0.03$ | **0.205** $\pm 0.03$ |

Lower = better (best in bold).

**Further insights:** Our framework leverages LLM-derived text to capture an induced text confounder during training. Hence, we acknowledge risks from LLM use, such as biases or data misrepresentation. We thus conduct experiments on subgroups in the datasets. (i) We split the datasets by gender into two groups $G_F$ and $G_M$ with females and males, respectively. (ii) We split the datasets by age into two groups $G_Y$ and $G_O$ as age before or above 45, respectively. Results are shown in Table 2. We observe minimal performance variation across subgroups, suggesting that the LLMs do not introduce significant subgroup-specific bias. Our method consistently outperforms baselines across different subpopulations.

• **Additional comparison with DR-learner:** Our TCA differs substantially from the standard DR-Learner. A simple off-the-shelf combination of our text-generation step with existing DR-learners would be biased. We additionally conduct experiments where we combine our text-generation pipeline with a standard DR learner [33] (referred to as TBE-DR). Importantly, we made the experiment fair: we used the same setup for the LLMs, etc. We report PEHE scores on two datasets, as shown in Table 5.

From the results, we can see our TCA outperforms TBE-DR by a large margin. This baseline uses the same underlying estimation technique as ours but lacks our framework for confounding adjustment. These results highlight that the core innovation lies in our confounding adjustment framework, not in the choice of DR itself. In general, while our approach leverages the doubly robust estimation to benefit from the doubly robust properties, it

Table 5: Additional comparison with DR-learner. Reported: mean $\pm$ standard deviation.

|  | IST | MIMIC-III |
|---|---|---|
| TBE-DR [33] | 0.187 $\pm 0.03$ | 0.266 $\pm 0.02$ |
| **TCA** (Ours) | **0.116** $\pm 0.02$ | **0.179** $\pm 0.02$ |

Lower = better (best in bold).

could also be easily adapted to use other meta-learners in the final step. The novelty lies in how we adjust for the text time confounding.

**Conclusion:** Our paper highlights the challenge of *inference time text confounding* for treatment effect estimation, where confounders that are fully observed during training are only partially observable through text at inference. This is common in any application of personalized medicine involving text input during clinical deployments, such as chatbots or medical LLMs for question answering. Our results show that our TCA framework can effectively yield reliable treatment effect estimates for personalized medicine, even when only limited textual descriptions are available at inference time.

# 7 Acknowledgments

This work has been supported by the German Federal Ministry of Education and Research (Grant: 01IS24082).

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

# A Extended related work

## A.1 Conditional average treatment effect (CATE)

Estimating the CATE has received a lot of attention in the machine learning literature (e.g.,[4, 6, 7, 28, 29, 39, 33, 41, 42, 43, 47, 54, 62]). A prominent approach to CATE estimation involves *meta-learners* – i.e., flexible strategies that decouple treatment effect estimation from the choice of base machine learning model. First systematized by [39], these methods have since been extended through theoretical analysis [34, 47] and improved pseudo-outcome constructions [33].

Existing CATE meta-learners can be categorized into: (a) one-step (plug-in) learners (indirect meta-learners) that output two regression functions from the observational data and then compute CATE as the difference in the potential outcomes (this is the strategy underlying the S- and T-learners); and (b) two-step learners (direct meta-learners/multi-stage direct estimators). These learners first compute nuisance functions to build a pseudo-outcome. In the second step, they obtain the CATE directly by regressing the input covariates on the pseudo-outcome. (Note that *pseudo-outcomes are not potential outcomes*). In terms of (b), existing methods fall largely into three broad classes: regression adjustment (RA), propensity weighting (PW), or doubly robust (DR) strategies.

The RA-learner uses the regression-adjusted pseudo-outcome in the second step, i.e.,

$$\tilde{Y}_{\text{RA},\hat{\eta}} = A\left(Y - \hat{\mu}_0(X)\right) + (1 - A)\left(\hat{\mu}_1(X) - Y\right) \tag{7}$$

The PW-leaner is inspired by inverse propensity-weighted (IPW) estimators, which is associated with pseudo-outcome, i.e.,

$$\tilde{Y}_{\text{PA},\hat{\eta}} = \left(\frac{A}{\hat{\pi}(X)} - \frac{1 - A}{1 - \hat{\pi}(X)}\right) Y \tag{8}$$

Doubly robust (DR) learners combine elements of RA and PW to mitigate their individual limitations. It is an extensions of augmented inverse probability weighting (AIPW) by constructing pseudo-outcomes using both propensity scores and outcome models. The pseudo-outcome is defined as:

$$\tilde{Y}_{\text{DR}} = \left(\frac{A}{\hat{\pi}(X)} - \frac{(1 - A)}{1 - \hat{\pi}(X)}\right) Y + \left[\left(1 - \frac{A}{\hat{\pi}(X)}\right) \hat{\mu}_1(X) - \left(1 - \frac{1 - A}{1 - \hat{\pi}(X)}\right) \hat{\mu}_0(X)\right]. \tag{9}$$

It can also be written in the residual form as

$$\tilde{Y}_{\text{DR}} = \hat{\mu}_1(X) - \hat{\mu}_0(X) + \frac{A \cdot (Y - \hat{\mu}_1(X))}{\hat{\pi}(X)} - \frac{(1 - A) \cdot (Y - \hat{\mu}_0(X))}{1 - \hat{\pi}(X)}. \tag{10}$$

As the latter is based on the doubly-robust AIPW estimator, it is hence unbiased if either propensity score or outcome regressions are correctly specified [33].

## A.2 Causal inference with text data

Causal inference with text data has emerged as an important research direction (e.g. [61, 31, 50, 45, 72, 44, 17, 32, 8, 70, 10, 13, 2]), with text variables serving multiple roles in causal frameworks.

Prior research has considered text to have various causal roles, with text serving as a treatment, mediator, outcome, or confounder. For example, previous works view text as treatment including [e.g., 58, 66, 64, 49, 44, 13]; works view text as mediator including[e.g., 72, 61, 17, 32]); and works view text as outcome including [e.g., 16, 57, 38]). There are many works that have explored text as a confounder [e.g., 31, 50, 45, 8, 70, 2]). [31] surveyed methods leveraging text to remove confounding and challenges. Some work attempts to mitigate confounding bias by viewing text features act as proxies for unobserved confounders [61, 50, 10, 2].

## A.3 NLP and language models in medical decision-making

Natural language processing (NLP) has emerged as an important tool in healthcare, enabling the extraction of meaningful information from unstructured clinical texts [e.g., 9, 3, 25, 73, 67, 52, 21, 23, 24]). Early work focused on using NLP to process clinical notes and symptom descriptions for some basic analysis and classification tasks, while more recent approaches leverage deep learning for

learning the representations, particularly transformer-based models like BERT and GPT [11, 25, 1, 12, 40]. Work in this area has focused on learning the representations; for instance,, [25] improved on previous clinical text processing methods and proposed a pretrained model on clinical notes to learn better representation for prediction. Other work used LLMs to impute the missing data in clinical notes [12].

Large language models (LLMs) have shown promise in medical applications, such as clinical text summarization, diagnosis, and treatment recommendations [59, 46, 56, 60, 12]. For example, [36] introduce a multi-agent framework that leverages LLMs to emulate the hierarchical diagnosis procedures, targeted for symptom classification and risk stratification tasks. However, unstructured text data often lacks explicit information about key confounders, which is critical for accurate causal inference. most NLP models, including LLMs, are designed for associative learning rather than causal reasoning [27, 26]. Their use in causal inference, particularly in addressing text confounding, remains underexplored.

### A.4 Counterfactual predictions under runtime confounding

Our setting is also related to [5], which refers to a scenario where historical data contains all the relevant information needed for decision-making. Only a subset of covariates $V \subseteq X$ is available to use at runtime. The unobserved set $X \setminus V$ should include all the hidden confounders at the runtime, denoted as $Z$. They show this can induce considerable bias in the resulting prediction model when the discarded features are significant confounders. However, this setting is different from ours. They require the set $V$ at runtime known and fixed. This is a strong and unrealistic assumption in our setting. Our setting allows $V$ to be an arbitrary subset of $X$. Moreover, their setting cannot handle the modality of data, which involves structured clinical tabular data at training time and only self-reported description-based text data at inference time.

# B Proof

## B.1 Pointwise confounding bias of the naïve baseline

**Lemma B.1** (Pointwise confounding bias of the naïve baseline). *Under inference time text confounding, assume that Assumption 3.1 holds. Then, for any $t \in \mathcal{T}$, the naïve estimator defined as $\tau_{\mathrm{naive}}^t(t) = \mathbb{E}[Y \mid A = 1, T = t] - \mathbb{E}[Y \mid A = 0, T = t]$ has pointwise bias with respect to the true conditional average treatment effect (CATE) $\tau^t(t)$, given by*

$$
\begin{aligned}
\mathrm{bias}(t) &= \tau_{\mathrm{naive}}^t(t) - \tau^t(t) \\
&= \left( \mathbb{E}[\mu_1^x(x) \mid A = 1, T = t] - \mathbb{E}[\mu_1^x(x) \mid T = t] \right) \\
&\quad - \left( \mathbb{E}[\mu_0^x(x) \mid A = 0, T = t] - \mathbb{E}[\mu_0^x(x) \mid T = t] \right),
\end{aligned} \tag{11}
$$

*where $\mu_a^x(x) = \mathbb{E}[Y \mid X = x, A = a]$.*

*Proof.* We give the proof by following [5].

By the consistency (Assumption 3.1(i)), we have

$$
\mathbb{E}[Y \mid A = a, T = t] = \mathbb{E}[Y(a) \mid A = a, T = t]. \tag{12}
$$

By the unconfoundedness (Assumption 3.1(ii)) and induced text generation mechanism (the independence of the noise $\epsilon$, i.e., $\epsilon \perp\!\!\!\perp Y(a) \mid X$), and by applying the law of iterated expectations, we obtain

$$
\mathbb{E}[Y(a) \mid A = a, T = t] = \mathbb{E}[\mu_a^x(x) \mid A = a, T = t]. \tag{13}
$$

Hence, the naïve estimator can be written as

$$
\tau_{\mathrm{naive}}^t(t) = \mathbb{E}[\mu_1^x(x) \mid A = 1, T = t] - \mathbb{E}[\mu_0^x(x) \mid A = 0, T = t]. \tag{14}
$$

The target CATE given $T$ is defined by

$$
\tau^t(t) = \mathbb{E}[Y(1) - Y(0) \mid T = t] = \mathbb{E}[\mu_1^x(x) - \mu_0^x(x) \mid T = t]. \tag{15}
$$

Subtracting the true CATE from the naïve estimator yields the bias

$$
\begin{aligned}
\mathrm{bias}(t) &= \tau_{\mathrm{naive}}^t(t) - \tau^t(t) \\
&= \left( \mathbb{E}[\mu_1^x(x) \mid A = 1, T = t] - \mathbb{E}[\mu_0^x(x) \mid A = 0, T = t] \right) - \mathbb{E}[\mu_1^x(x) - \mu_0^x(x) \mid T = t] \\
&= \left( \mathbb{E}[\mu_1^x(x) \mid A = 1, T = t] - \mathbb{E}[\mu_1^x(x) \mid T = t] \right) \\
&\quad - \left( \mathbb{E}[\mu_0^x(x) \mid A = 0, T = t] - \mathbb{E}[\mu_0^x(x) \mid T = t] \right).
\end{aligned} \tag{16}
$$

$\square$

*Remark* B.2 (Non-zero bias of the naïve estimator). The bias of the naïve estimator remains non-zero under Assumption 3.1 and $Y(0), Y(1) \not\perp\!\!\!\perp A \mid T$.

In the context of inference time text confounding, when $T$ does not capture all the necessary confounding information in $X$ for potential outcome, the naïve estimator is necessarily biased, as

$$
\mathbb{E}[\mu_a^x(X) \mid A = a, T = t] \neq \mathbb{E}[\mu_a^x(X) \mid T = t]. \tag{17}
$$

## B.2 Identifiablity of $\tau^t(t)$ by adjusting for inference time text confounding

**Lemma B.3** (Identifiablity of $\tau^t(t)$). *For any $t \in \mathcal{T}$, under Assumption 3.2, $\tau^t(t)$ can be identified through $\tau^x(x)$ via*

$$\tau^t(t) = \mathbb{E}\left[\tau^x(X) \mid T = t\right]. \tag{18}$$

*Proof.* By definition, the CATE with respect to the induced text confounder is

$$\tau^t(t) = \mathbb{E}[Y(1) - Y(0) \mid T = t]. \tag{19}$$

Applying the law of iterated expectations, we can condition on the true confounder $X$:

$$\tau^t(t) = \mathbb{E}[\mathbb{E}[Y(1) - Y(0) \mid T = t, X] \mid T = t]. \tag{20}$$

Under Assumption 3.2, conditional on $X$, the variable $T$ does not provide any additional information about the potential outcomes, we have

$$\mathbb{E}[Y(1) - Y(0) \mid X, T = t] = \mathbb{E}[Y(1) - Y(0) \mid X] = \tau^x(x). \tag{21}$$

Thus, we obtain

$$\tau^t(t) = \mathbb{E}\left[\tau^x(x) \mid T = t\right]. \tag{22}$$

$\square$

### B.3 Double robustness property of the estimator

**Corollary B.4** (Double robustness property of the estimator)**.** *The estimator satisfies the double robustness property by construction: $\hat{\tau}^t(t)$ converges to the true $\tau^t(t)$ if either (i) the response surface estimates $\hat{\mu}_a^x$ are consistent, or (ii) the propensity score estimate $\hat{\pi}^x$ is consistent. This ensures validity even under potential model misspecification.*

Here, we give the proof of the Corollary B.4, which states that our estimator satisfies the double robustness property. Formally, we have the response functions defined as $\mu_a^x(x) = \mathbb{E}[Y \mid X = x, A = a]$, and $\pi^x(x) = \mathbb{P}(A = 1 \mid X = x)$ as the true propensity score. Then, the true conditional average treatment effect (CATE) is

$$\tau^x(x) = \mu_1^x(x) - \mu_0^x(x). \tag{23}$$

Our doubly robust pseudo-outcome is given as

$$\tilde{Y} = \hat{\mu}_1^x(x) - \hat{\mu}_0^x(x) + \frac{A}{\hat{\pi}^x(x)}\left(Y - \hat{\mu}_1^x(x)\right) - \frac{1 - A}{1 - \hat{\pi}^x(x)}\left(Y - \hat{\mu}_0^x(x)\right), \tag{24}$$

where $\hat{\mu}_a^x(x)$ and $\hat{\pi}^x(x)$ are estimators of $\mu_a^x(x)$ and $\pi^x(x)$, respectively.

We now show that, under the Assumption 3.1, if either

1. the outcome models are correctly specified, i.e., $\hat{\mu}_a^x(x) = \mu_a^x(x)$ for $a \in \{0, 1\}$, or
2. the propensity score model is correctly specified, i.e., $\hat{\pi}^x(x) = \pi^x(x)$,

it follows that

$$\mathbb{E}\left[\tilde{Y} \mid X = x\right] = \tau^x(x). \tag{25}$$

*Proof.* We give the proof following [33]. We treat the two cases separately.

**Case 1: Correct outcome models.**

Assume that $\hat{\mu}_a^x(x) = \mu_a^x(x)$ for $a \in \{0, 1\}$. Then, the pseudo-outcome reduces to

$$\tilde{Y} = \mu_1^x(x) - \mu_0^x(x) + \frac{A}{\hat{\pi}^x(x)}\left(Y - \mu_1^x(x)\right) - \frac{1 - A}{1 - \hat{\pi}^x(x)}\left(Y - \mu_0^x(x)\right). \tag{26}$$

Taking the expectation conditional on $X = x$ gives

$$\begin{aligned} \mathbb{E}\left[\tilde{Y} \mid X = x\right] = {} & \mu_1^x(x) - \mu_0^x(x) \\ & + \mathbb{E}\left[\frac{A}{\hat{\pi}^x(x)}\left(Y - \mu_1^x(x)\right) \,\middle|\, X = x\right] \\ & - \mathbb{E}\left[\frac{1 - A}{1 - \hat{\pi}^x(x)}\left(Y - \mu_0^x(x)\right) \,\middle|\, X = x\right]. \end{aligned} \tag{27}$$

We have

$$\mathbb{E}\left[Y - \mu_1^x(x) \mid X = x, A = 1\right] = 0 \quad \text{and} \quad \mathbb{E}\left[Y - \mu_0^x(x) \mid X = x, A = 0\right] = 0, \tag{28}$$

and, as both expectation terms vanish, yielding

$$\mathbb{E}\left[\tilde{Y} \mid X = x\right] = \mu_1^x(x) - \mu_0^x(x) = \tau^x(x). \tag{29}$$

**Case 2: Correct propensity score model.** Now, assume that $\hat{\pi}^x(x) = \pi^x(x)$, while allowing for misspecification of the outcome models. We define the errors

$$\delta_a(x) = \hat{\mu}_a^x(x) - \mu_a^x(x), \quad a \in \{0, 1\}, \tag{30}$$

so that $\hat{\mu}_a^x(x) = \mu_a^x(x) + \delta_a(x)$. Then, the pseudo-outcome can be written as

$$\begin{aligned} \tilde{Y} = {} & \left[\mu_1^x(x) + \delta_1(x)\right] - \left[\mu_0^x(x) + \delta_0(x)\right] \\ & + \frac{A}{\pi^x(x)}\left(Y - \mu_1^x(x) - \delta_1(x)\right) \\ & - \frac{1 - A}{1 - \pi^x(x)}\left(Y - \mu_0^x(x) - \delta_0(x)\right). \end{aligned} \tag{31}$$

Taking the conditional expectation given $X = x$, we have

$$
\begin{aligned}
\mathbb{E}\left[\tilde{Y} \mid X = x\right] = {} & \mu_1^x(x) - \mu_0^x(x) + \delta_1(x) - \delta_0(x) \\
& + \frac{1}{\pi^x(x)} \mathbb{E}\left[A\left(Y - \mu_1^x(x) - \delta_1(x)\right) \mid X = x\right] \\
& - \frac{1}{1 - \pi^x(x)} \mathbb{E}\left[(1 - A)\left(Y - \mu_0^x(x) - \delta_0(x)\right) \mid X = x\right].
\end{aligned}
\tag{32}
$$

Noting that $\mathbb{E}[Y \mid X = x, A = a] = \mu_a^x(x)$, it follows that

$$
\mathbb{E}\left[A\left(Y - \mu_1^x(x)\right) \mid X = x\right] = 0 \quad \text{and} \quad \mathbb{E}\left[(1 - A)\left(Y - \mu_0^x(x)\right) \mid X = x\right] = 0.
\tag{33}
$$

Furthermore, by linearity,

$$
\mathbb{E}\left[A\,\delta_1(x) \mid X = x\right] = \delta_1(x)\,\pi^x(x), \quad \mathbb{E}\left[(1 - A)\,\delta_0(x) \mid X = x\right] = \delta_0(x)\,(1 - \pi^x(x)).
\tag{34}
$$

Thus, we have

$$
\frac{1}{\pi^x(x)} \mathbb{E}\left[A\left(Y - \mu_1^x(x) - \delta_1(x)\right) \mid X = x\right] = -\delta_1(x),
\tag{35}
$$

and

$$
-\frac{1}{1 - \pi^x(x)} \mathbb{E}\left[(1 - A)\left(Y - \mu_0^x(x) - \delta_0(x)\right) \mid X = x\right] = \delta_0(x).
\tag{36}
$$

Substituting these back, we obtain

$$
\begin{aligned}
\mathbb{E}\left[\tilde{Y} \mid X = x\right] &= \mu_1^x(x) - \mu_0^x(x) + \delta_1(x) - \delta_0(x) - \delta_1(x) + \delta_0(x) \\
&= \mu_1^x(x) - \mu_0^x(x) \\
&= \tau^x(x).
\end{aligned}
\tag{37}
$$

$\square$

# C    Implementation details

**Text generation:** Given structured clinical confounders $X$, we generate the induced text confounder $\tilde{T}$ using GPT-4o mini through the OpenAI API. For each patient, we construct prompts by templating the key features of $X$ into natural language constraints. Generated texts are approximately $150 - 200$ tokens.

**Text embedding:** We convert generated text $\tilde{T}$ into dense vectors using pretrained BERT (bert-base-uncased) from HuggingFace Transformers. For each narrative, we compute token embeddings from the final transformer layer and apply mean pooling across tokens, yielding fixed-dimensional representations $\phi\left(\tilde{t}_i\right) \in \mathbb{R}^{768}$.

**Model architecture and training:** Propensity scores $\hat{\pi}^x(x)$ are estimated via logistic regression The CATE predictor is a 3-layer MLP with ReLU activation, batch normalization, and 0.3 dropout. We use Adam optimizer with learning rate $5e - 5$. Models are trained for 100 epochs on IST and 150 on MIMIC-III with a batch size of $512$. We apply label smoothing $\alpha = 0.1$ and gradient clipping (max norm=1.0)

**Reported time:** Experiments were carried out on 2 GPUs (NVIDIA A100-PCIE-40GB) with Intel Xeon Silver 4316 CPUs. Constructing each dataset took approximately 50 hours. Training our `TCA` takes around 10 minutes on average, comparable to the baseline training times.

**Baselines:** We follow the implementation from `https://github.com/AliciaCurth/CATENets/tree/main` for most of the CATE estimators, including S-Net [39], T-Net [39], TARNet [54], CFRNet [54]. Regarding the proxy-based methods [61, 2], we follow the implementation from `https://github.com/rpryzant/causal-bert-pytorch` and the implementation from `https://github.com/jacobmchen/proximal_w_text`.

# D  Dataset

## D.1  The International Stroke Trial database

The International Stroke Trial (IST) [53] was conducted between 1991 and 1996. It is one of the largest randomized controlled trials in acute stroke treatment. The dataset comprises $19,435$ patients from 467 hospitals across 36 countries, enrolled within 48 hours of stroke onset. The treatment assignments include (trial aspirin allocation and trial heparin allocation with levels for medium dose, low dose, and no heparin). The outcome is whether a patient will experience stroke again. Covariates include age, sex, presence of atrial fibrillation, systolic blood pressure, infarct visibility on CT, prior heparin use, prior aspirin use, and recorded deficits of different body parts.

## D.2  MIMIC-III dataset

The Medical Information Mart for Intensive Care (MIMIC-III) [30] is a large, single-center database comprising information relating to patients admitted to critical care units at a large tertiary care hospital. MIMIC-III contains $38,597$ distinct adult patients. We follow the standardized preprocessing pipeline [63] of the MIMIC-III dataset. We use demographic variables such as gender and age, along with clinical variables including vital signs and laboratory test results upon admission, as confounders. These include glucose, hematocrit, creatinine, sodium, blood urea nitrogen, hemoglobin, heart rate, mean blood pressure, platelets, respiratory rate, bicarbonate, red blood cell count, and anion gap, among others. The binary treatment is mechanical ventilation. The outcome is the number of days a patient needs to stay in the hospital.

We follow previous work [39, 54, 6, 7, 33] to simulate treatment and outcome as follows

$$\begin{cases} A \sim \text{Bernoulli}(0.5), \\ Y = \sigma(\beta^T X + \gamma_1 A + \gamma_2 A^2 + \gamma_3 \sin(A)) + (\delta^T X) A + \epsilon, \end{cases} \tag{38}$$

where $\beta$, $\delta$, $\gamma_1$, $\gamma_2$, and $\gamma_3$ are fixed coefficients; $\sigma$ refers to the sigmoid function $\sigma(z) = \frac{1}{1+e^{-z}}$ and $\epsilon$ is a noise term.

## D.3  Experiments with varying confounder strengths

To evaluate the performance of TCA under varying confounder strengths, we modify the data-generating process by introducing scaling parameters that control the influence of the confounders $X$ on the outcome $Y$ and/or the treatment assignment $A$.

Specifically, for the outcome model, we use a scaling parameter $\eta \geq 0$, which controls the strength of the confounding effect via the interaction term $(\delta^\top X)A$. Setting $\eta = 0$ corresponds to no confounding from this term, whereas larger values of $\eta$ induce stronger confounding. Alternatively, to also vary the confounding in the treatment assignment, we can generate $A$ via a logistic model

$$\Pr(A = 1 \mid X) = \sigma\left(\kappa \xi^\top X\right), \tag{39}$$

where $\kappa \geq 0$ regulates the influence of $X$ on $A$, $\xi$ is a parameter vector, and $\sigma(z) = \frac{1}{1+e^{-z}}$ denotes the sigmoid function.

In our experiments, we vary $\eta$ (and/or $\kappa$) over a predetermined grid (e.g., $\eta \in \{0.5, 1.0, 1.5\}$) to simulate different levels of confounder strength. As the confounding strength increases, we expect that the naïve text-based estimator (which relies solely on the generated text surrogate $\tilde{T}$ and assumes unconfoundedness) will incur increasing bias due to unadjusted residual confounding. In contrast, our TCA framework leverages the full confounder $X$ for robust nuisance function estimation and employs a doubly-robust regression that conditions on $\tilde{T}$. Therefore, our method should remain robust, which should thus widen the performance gap in favor of TCA as the confounder strength increases.

*Prediction performance under varying confounder strength:* When the true confounder strength is low, both our TCA framework and the naïve text-based estimators are expected to yield comparable CATE estimates since residual confounding is minimal. However, as the confounding strength increases, the naïve estimators which rely solely on the generated text surrogate $\tilde{T}$ and assume unconfoundedness will incur greater bias due to unadjusted residual confounding. In contrast, our method leverages the

full confounder $X$ for robust nuisance function estimation and employs a doubly-robust regression that conditions on $\tilde{T}$. As a result, TCA exhibits significantly improved performance under strong confounding, leading to an increasing performance gap between our method and the naïve baselines as confounder strength increases.

## D.4 Experiments with varying prompt strategies

We test three prompt families to generate $\tilde{T}$.

**(i) Factual prompt:**

> **Template example:**
>
> ```
> Transfer this patient information into a paragraph of text.  We have
> the patient clinical information including gender, age, glucose,
> hematocrit, creatinine, sodium, blood urea nitrogen, hemoglobin,
> heart rate, mean blood pressure, platelets, respiratory rate,
> bicarbonate, red blood cell count, and anion gap.  The values are
> M,41,129.0,45.2,1.2,140.0,16.0,15.4,84.0,103.4,258.0,12.0,30.0,5.46,13.0.
> ```

> **Example output:**
>
> ```
> The patient is a 41-year-old male with a glucose level of 129.0 mg/dL.
> His hematocrit is 45.2%, and creatinine is measured at 1.2 mg/dL.
> The sodium level is 140.0 mEq/L, while the blood urea nitrogen is
> 16.0 mg/dL. He has a hemoglobin level of 15.4 g/dL, a heart rate of
> 84.0 beats per minute, and a mean blood pressure of 103.4 mmHg.  The
> patient's platelet count is 258.0 x 10⁹/L, with a respiratory rate of
> 12.0 breaths per minute.  His bicarbonate level is 30.0 mEq/L, and his
> red blood cell count is 5.46 million cells per μL , resulting in an
> anion gap of 13.0 mEq/L.
> ```

**(ii) Narrative prompt:**

> **Template example:**
>
> ```
> Write a detailed clinical narrative for a patient with these
> features:  gender, age, glucose, hematocrit, creatinine,
> sodium, blood urea nitrogen, hemoglobin, heart rate, mean
> blood pressure, platelets, respiratory rate, bicarbonate,
> red blood cell count, and anion gap.  The values are
> F,65,214.0,29.55,11.7,138.0,62.0,9.8,83.0,77.0,315.0,13.0,15.0,3.44,23.0.
> ```

**Example output:**

The patient, a 65-year-old female, presents with significant
hyperglycemia, indicated by a glucose level of 214.0 mg/dL, which
suggests poor blood sugar control.  Other vital signs reflect some
abnormalities; her hematocrit is elevated at 29.55%, indicating
possible dehydration or underlying blood issues.  The creatinine level
at 11.7 mg/dL is concerning for potential kidney dysfunction, while
sodium is within the normal range at 138.0 mmol/L. The blood urea
nitrogen is raised at 62.0 mg/dL, which might further point to renal
impairment or dehydration.  Her heart rate is slightly elevated at
83.0 beats per minute, and mean blood pressure is on the lower side
at 77.0 mmHg, which could suggest compromised cardiovascular status
or shock.  The platelet count is high at 315.0 x10$^9$/L, and respiratory
rate and bicarbonate levels appear stable.  Overall, the patient is
likely feeling unwell, with symptoms potentially related to high blood
sugar levels and possible kidney issues, and she may require further
evaluation and management to address these concerning lab results.
Access to healthcare facilities for comprehensive diagnostic testing
and treatment is essential at this stage.

**(iii) Symptom-focused prompt:**

**Template example:**

Here is a patient current information (if this patient
can use diagnostic equipment or has access to healthcare
facilities):  The variables include gender, age, glucose,
hematocrit, creatinine, sodium, blood urea nitrogen, hemoglobin,
heart rate, mean blood pressure, platelets, respiratory rate,
bicarbonate, red blood cell count, and anion gap.  The values are
M,76,198.56,27.03,2.6,142.67,44.2,8.72,143.0,101.56,233.6,16.0,16.33,
2.88,20.67.  How would this patient describe his/her symptoms via text
with just partial diagnostic measurements?

**Example output:**

I've been feeling really unwell lately.  I'm a 76-year-old man, and
I've noticed I've been more tired than usual, with a lack of energy
that makes it difficult to engage in my daily activities.  There's
this constant feeling of thirst that I can't shake off, and I seem
to be going to the bathroom more often.  My heart feels like it's
racing at times, and I get short of breath, especially when I try
to move around or do any little tasks.  Sometimes, I experience a
strange sensation in my head, almost like dizziness or confusion.
I've also noticed some swelling in my legs.  It worries me, and I feel
frustrated not being able to enjoy my usual routine or activities.  I
just want to feel better and get back to my normal life.

$\Rightarrow$ **Expected performance differences between prompting technqiues:** Since our method is designed to correct for the residual confounding by using the full confounders $X$, it is inherently more robust to variations in prompt quality. In contrast, the naïve baseline directly relies on $\tilde{T}$ for adjustment and therefore is more sensitive to the quality and informativeness of the text. With low-quality prompts (i.e., the factual prompt), both methods might perform similarly poorly, though our method is still expected to have an advantage due to its use of $X$. With high-quality prompts (i.e., the narrative or symptom-focused), the performance of our method is expected to improve substantially because it can fully exploit the richer text representations to accurately map onto the

treatment effects. Meanwhile, the naïve baseline remains biased due to its inability to adjust for the missing confounder components from $X$.

# E   Additional acknowledge

We have access to ChatGPT Edu through our university, which is SOC 2 Type II compliant (`https://openai.com/index/introducing-chatgpt-edu/`). This gives us enhanced privacy controls not available in the standard API, free version, or even ChatGPT Team.

As part of this setup, we have applied for a Business Associate Agreement (BAA) (`https://help.openai.com/en/articles/8660679-how-can-i-get-a-business-associate-agreement-baa-with-openai-for-the-api-services`) to gain access to Zero Data Retention (ZDR). With ZDR enabled, the `store` parameter for `/v1/responses` and `v1/chat/completions` will always be treated as `false` `https://platform.openai.com/docs/guides/your-data#zero-data-retention`. In this setup, OpenAI does not retain any request or response data after processing, meaning there is no storage, no logging, and, more importantly, no human review. This configuration provides a privacy approach that is functionally equivalent to the one recommended by PhysioNet `https://physionet.org/news/post/gpt-responsible-use`, where Azure OpenAI is used with human review explicitly disabled. We confirm that our analysis is thus HIPAA-compliant and thus in line with the PhysioNet rules.

# F    Further discussion

Our framework leverages LLM-derived text to capture an induced text confounder during training. Hence, we acknowledge risks from LLM use, such as biases or data misrepresentation. We thus advocate for a rigorous validation of LLM outputs with domain experts to ensure reliability and safety, particularly in high-stakes contexts in medicine. Further, LLMs should be used transparently and ethically [15], with safeguards against amplifying societal biases. Future work should explore integrating uncertainty quantification to further increase trust in our TCA during real-world deployment.

Technically, our method relies on the fact that LLMs generate faithful representations, meaning that generated text $\tilde{T}$ and text embedding preserve semantic relationships. Importantly, such challenges are not unique to our method; any baseline relying on text generation or embeddings would similarly be impacted by low-quality text or suboptimal embeddings. This highlights the broader importance of leveraging robust LLMs and embedding techniques in text-based causal inference tasks.

Notwithstanding, our proposed framework has also benefits for making reliable inferences. By addressing inference time text confounding, our framework focuses on an important but overlooked setting where the true text confounders are missing at inference time. Such a setting is common in telemedicine (e.g., when LLMs are used for making treatment recommendations where access to physicians is often limited or where access to diagnostic infrastructure is lacking). Here, our framework can help reduce bias in CATE estimation and can thus help improve personalized medicine.

