# OpenReview forum: "LLM-Driven Treatment Effect Estimation Under Inference Time Text Confounding"
_NeurIPS.cc/2025/Conference — NeurIPS 2025 poster_

### Official Review · Reviewer_zRnF · 2025-06-09

**Clarity:** 3
**Significance:** 2
**Originality:** 2
**Rating:** 3
**Confidence:** 4

**Summary:**

This paper investigate the CATE estimation problem with text information during inference time while the complete confounder is available during training time. To bridge this gap, the authors first train a doubly robust estimator using the training dataset. Then, the author construct text description of the confounders in the training data. Based on the generated text, the algorithm constructs a mapping from the text to the estimated treatment effect. Experiemental results on International Stroke Trial and MIMIC-III demonstrate the effectiveness of the method.

**Questions:**

1. The Assumption 3.2 is disjointed with the other parts of the paper. How to understand the noise variable in the text generation mechanism?

2. How to obtain the text of the test set in MIMIC-III and IST?

3. The proposed TCA method use doubly robust method as the estimation. However, the baselines does not use doubly robust method. For fair comparision, I suggest to supplment the TBE-Doubly Robust as the baseline.

**Ethical Concerns:**

["NO or VERY MINOR ethics concerns only"]

**Limitations:**

The authors present the limitations in Appendix G. As the authors claimed, the quality of the generated text and representation is important to the effectiveness of the method. And the other point is also important. Since the generated text contains less information than the confounder variable $X$, the single value of $T$ refers to multiple value of $X$. However, this fact is not considered in the method and a deterministic mapping is learned. Therefore, intergrating uncertainty quantification is necessary to this method.

**Paper Formatting Concerns:**

No paper formatting concerns.

**Quality:**

2

**Strengths And Weaknesses:**

The paper is well written and presented. The experimental results demonstrate the significant advancement of the proposed method. However, I think some weaknesses still exist as follows:

1. The motivation is somehow weak. Because the text is much easier to obtain than complete measurement of confounders, it is not realistic that during training time, the text is unavailable.

2. The novelty is not sufficient. The estimation method is the off-the-shelf doubly robust method. The incremental part is only the text generation and mapping function.

3. The text generation based on LLM may be misaligned with the real text collection. Therefore, the rationality of the mapping function from the text to treatment effect is not convincing.

---

> ### Author Rebuttal · Authors · 2025-07-31
>
> Thank you for your constructive review and your helpful comments!
>
>
> ### Response to Weakness
>
> **W1**
>
> We appreciate the reviewer’s concern, but respectfully clarify that our setting—where text is unavailable during training but present at inference—is both realistic and increasingly common, especially in the medical domain.
>
> In many real-world healthcare applications, it is common that structured data (e.g., demographics, vitals, labs) is reliably recorded and readily available during training, particularly in randomized controlled trials or publicly released clinical datasets. In contrast, unstructured text is often sparse or inaccessible, or incomplete, inconsistently formatted, and lacks standardization.
> Structured tabular data are carefully recorded and quality checked in clinical datasets, making them easily available. This is typically driven by regulatory need and approval for medical devices, where only high-quality information is preferred over self-reported texts. As a result, unstructured text is either unavailable, inconsistently documented. Models are often built on well-structured datasets from electronic health records or clinical trials.
>
> Even when some text is present during training (e.g., clinical notes), it is often incomplete, inconsistently formatted, and often differs substantially in style and content from inference-time text (e.g., from a different distribution and with language style). For example, training-time notes are authored by clinicians, while inference-time inputs in telemedicine, chatbots, or symptom-reporting apps are often patient-generated. Such free-text symptom descriptions often fail to cover the full set of confounding variables.
>
> Our setting, where models are trained on structured data but must make inferences on text data is both practically motivated and clinically relevant. Our work is the first to formally characterize the challenge of inference-time text confounding and to propose a concrete solution that is theoretically principled and empirically validated in such real-world conditions.
>
> **W2**
>
> We would like to clarify that our method is **not** the off-the-shelf doubly robust method. Our TCA **differs substantially** from the standard DR-Learner. _A simple off-the-shelf combination of our text-generation step with existing DR-learners would be biased_.
>
> Our main contribution includes
>
> (1) Problem formalization: We are the first to **formally define the inference-time text confounding problem**, which is distinct from standard causal inference settings. We show that naïve applications of existing methods (including DR learners) are biased in this setup, and we **derive their pointwise confounding bias**.
>
> (2) Theoretical contribution: We propose an estimation framework that connects the CATE $\tau^x(x)$ w.r.t true confounder $X$ to the text-based CATE $\tau^t(t)$, and we prove identifiability and unbiasedness of our estimator under this framework. This is not a trivial extension of existing DR methods.
>
> (3) Modular estimation strategy: While our approach leverages the doubly robust estimation to benefit from the doubly robust properties, it could also be easily adapted to use other meta-learners in the final step.
>
> (4) Empirical validation: We further **conduct new experiments** where we combine our text-generation pipeline with a standard DR learner (referred to as **TBE-DR**). Importantly, we made the experiment _fair_: we used the same setup for the LLMs, etc. We report PEHE scores on two datasets.
>
> | Dataset | TBE-DR | TCA (ours) |
> |---------|--------|------------|
> | IST     | 0.187  | 0.116      |
> | MIMIC   | 0.266  | 0.179      |
>
> => **Our TCA outperforms TBE-DR by a large margin**.
>
> This baseline uses the **same underlying estimation technique** as ours but lacks our framework for confounding adjustment. These results highlight that **the core innovation lies in our confounding adjustment framework**, not in the choice of DR itself.
>
> **W3**
>
> Thank you for raising this important point. We acknowledge that LLM-generated text may not perfectly align with real-world text distributions. Our method is explicitly designed to tolerate such mismatches.
> To assess robustness, we conducted domain shift experiments by varying the LLM prompt strategies across datasets (Figure 3(b)). Despite this shift, our method **consistently outperforms** all baselines, demonstrating **robustness** to the **distribution shift between text data**.
>
> ### Response to Questions
>
> **Q1**
>
> Thank you for your question. Assumption 3.2 defines the data-generating mechanism of the induced text confounder $T$, which is generated from the true confounder $X$ as $T = h(X, \epsilon)$, where the noise variable satisfies $\epsilon \perp Y(a) \mid X$.
> This assumption formalizes the intuition that $T$ is a noisy (i.e., stochastic) transformation of the underlying true confounder $X$,  which reflects the nature of real-world text generation in clinical settings. For example:
>
> (1) Doctor’s notes or patient self-descriptions are often incomplete, informal, and subject to variability (noise) such as different writing styles, which are, however, independent of the outcome of interest.
>
> (2) LLM-generated text, even when conditioned on structured inputs, introduces randomness (e.g., due to sampling variability or temperature settings, or seed, etc.).
>
> The noise variable $\epsilon$ models this stochasticity in text generation, and allows diverse formats of the text description of the true confounder.
>
> Importantly, We do not require that $T$ preserves all relevant information from $X$. It also fits in applications where $T$ often does not capture all the necessary confounding information in $X$ for potential outcomes. This aligns with the imperfect, noisy, and incomplete nature of text data in practice. Instead, $T$ may contain only partial information about $X$. As $T$ is induced by $X$, the useful information encoded by $T$ should be included within the information contained in $X$ for predicting the outcome.
>
> Besides, the independence condition $\epsilon \perp Y(a) \mid X$ ensures that:
> Once we condition on $X$, the residual information in $T$ does not provide any additional information about the potential outcomes.
>
> This assumption is more for technical reasons to ensure identifiability, that is, identifiability of $\tau^t(t)$ by adjusting for inference time text confounding, which allows us to express the CATE under text ($\tau^t(t)$) as a conditional expectation of the true CATE ($\tau^x(x)$), i.e., $\tau^t(t)=\mathbb{E}\left[\tau^x(X) \mid T=t\right]$.
> as shown in Lemma 4.3 and Appendix B.2. The assumption thus allows us to use our two-stage procedure for adjusting inference-time confounding when only $T$ is observed at test time.
>
> **Q2**
>
> The text for our experiments on top of  MIMIC-III and IST is generated by LLMs to simulate text with self-reported symptoms. In the datasets, there is no text available that captures such symptom descriptions. Hence, we use LLMs to generate textual confounders $\tilde{T}$ from structured clinical covariates $X$.  For each patient, we construct prompts by templating the key features of $X$ into natural language constraints. The generated texts are about 150–200 tokens long. We split the data into 80% train and 20% test.
>
> **Q3**
>
> Thank you for the suggestion. We **added the TBE-Doubly Robust (TBE-DR) learner as another baseline**. The corresponding results are reported above under W2.
>
> ### Response to Limitations
>
> Thank you for the thoughtful comment.
>
> Assumption 3.2 formalizes the intuition that $T$ is a noisy (i.e., stochastic) transformation outcome of the underlying true confounder $X$.   We do not assume the mapping from $X$ to $T$ is deterministic, one-to-one, or invertible. Nor do we require $X$ to be reconstructed from $T$ or that $T$ preserves all confounding information. A single $T$ may correspond to multiple possible values of $X$.
>
> It is true that the generated text $T$ often contains less information than the true confounder $X$, and this is **precisely why our framework is needed**—to adjust for inference-time confounding when only $T$ is available to use at inference time.
>
> To study this systematically, we conduct experiments under varying levels of information preserved in $T$, which we refer to as high-info, medium-info, and low-info settings, based on how much information from $X$ is retained in the text.
>
> We also appreciate the reviewer’s suggestion on incorporating uncertainty quantification.
> We thus measure the discrepancy between estimated $\hat{\tau}^t(t)$ and ground-truth $\tau^x(x)$. This discrepancy captures the uncertainty introduced by the loss of information in the transition from $X$ to $T$, introducing variability in $\tau^t(t)$ relative to the true effect $\tau^x(x)$. For reasons of space, we focus on TBE-DRlearner as it is the strongest baseline. We report results on IST and MIMIC, respectively. These results demonstrate that our framework is more robust to uncertainty.
>
> |Info Level|Method|$\mathbb{E}[\hat{\tau}^t(t)-\tau^x(x)]$|$\sqrt{\mathbb{E}\left[\left(\hat{\tau}^t(t) - \tau^x(x)\right)^2\right]}$|
> |----------|------|-------------------------------|--------------------------------------|
> |High-info|TBE-DR|0.0320|0.134|
> |         |Ours (TCA)|**0.0106**|**0.085**|
> |Medium-info|TBE-DR|0.0302|0.160|
> |           |Ours (TCA)|**0.0124**|**0.097**|
> |Low-info|TBE-DR|0.0431|0.187|
> |        |Ours (TCA)|**0.0142**|**0.116**|
>
>
> |Info Level|Method|$\mathbb{E}[\hat{\tau}^t(t)-\tau^x(x)]$|$\sqrt{\mathbb{E}\left[\left(\hat{\tau}^t(t) - \tau^x(x)\right)^2\right]}$|
> |----------|------|-------------------------------|--------------------------------------|
> |High-info|TBE-DR|0.0450|0.210|
> |         |Ours (TCA)|**0.0148**|**0.126**|
> |Medium-info|TBE-DR|0.0560|0.240|
> |           |Ours (TCA)|**0.0183**|**0.145**|
> |Low-info|TBE-DR|0.0673|0.266|
> |        |Ours (TCA)|**0.0206**|**0.179**|

---

> > ### Author Response · Authors · 2025-08-06
> >
> > Thank you again for your effort in reviewing this paper and your constructive comments! If the response addresses your concerns, we would greatly appreciate it if you would raise your score accordingly. Thank you very much for reviewing our paper!

---

> > ### Comment · Reviewer_zRnF · 2025-08-07
> >
> > Thanks for your reply. The response address most of my concerns. Since I still think the novelty is kind of weak, I decide to maintain my score. However, I would have no objection if the AC decides to accept this paper.

---

> > > ### Author Response · Authors · 2025-08-07
> > >
> > > Thank you again for taking the time to review our paper. We are glad that our response addressed most of your concerns.
> > >
> > >
> > > Regarding the point on novelty, we respectfully emphasize that our work differs substantially from the prior works. We believe our work makes a meaningful contribution by formalizing the inference-time text confounding problem and proposing a novel framework tailored to give unbiased CATE estimation. Importantly, the significance of our approach is supported by **strong** and state-of-the-art **empirical** performance. Our method outperforms baselines by a large margin.
> > >
> > >
> > > We truly appreciate your thoughtful feedback and your openness to the **acceptance** of our paper.

---

### Official Review · Reviewer_kWL2 · 2025-06-25

**Clarity:** 3
**Significance:** 3
**Originality:** 3
**Rating:** 4
**Confidence:** 5

**Summary:**

This paper proposes a method for causal effect estimation when confounder values are observed as textual description at test/inference time. The proposed problem setup and the solution proposed are novel and are first of its kind.The proposed method has real-world use cases especially in healthcare domains. The proposed idea identifies conditional average treatment effect conditional on textual confounders using the conditional average treatment effect conditional on actual/numerical values of confounders. Experimental results presented show improvements over naive baselines.

**Questions:**

Please see the weaknesses section and provide your responses.

**Ethical Concerns:**

["NO or VERY MINOR ethics concerns only"]

**Final Justification:**

All my concerns are addressed and I hope the assumption, which is strong, that all covariates are assumed to be confounders, is made more explicit in the revision. I will still keep my positive score despite the strong assumption because of the immediate real-world applicability of the work.

**Limitations:**

Yes

**Quality:**

3

**Strengths And Weaknesses:**

Strengths:

1. The proposed problem setup of causal effect estimation under test time text confounders is novel, and has real-world significance.
2. The proposed solution is intuitive and theoretically grounded.
3. The experimental results show good performance compared to baselines in all experimental setups.

Weaknesses:

1. In the abstract the following line overstates the contributions as it is well known fact.

"We show that the discrepancy between the data available during training time and inference time can lead to biased estimates of treatment effects."

2. A major issue with the current problem setup is that all covariates X are considered to be confounders. This holds very rarely in real-world. Many variables should not be used for adjustment as there may be covariates that are descendants of the treatment variable.

3. The significance of Lemma 4.1 is unknown because it is obvious that the bias of naive estimator is positive.

4. Finally, the proposed method tries to estimates $\tau^t(t)$ instead of $\tau^x(x)$. In my opinion, identifying $\tau^x(x)$ at test time would be even more interesting. At least comparing the estimated values of  $\tau^t(t)$ and $\tau^x(x)$ in "in-sample" data and checking that they do not deviate much from each other would be a great addition to the paper and validates the idea of estimating $\tau^t(t)$.

---

> ### Author Rebuttal · Authors · 2025-07-31
>
> Thank you for your positive review and your helpful comments! We are very happy to answer your questions and improve our paper based on your suggestions. Below, we address your comments in detail.
>
> ### Response to Weakness
>
> **Response to W1**
>
> Thank you for the suggestion. We agree that the original phrasing in the abstract may have overstated what is already known. Our intention was to highlight that we formally define this issue under inference-time text confounding, which, to the best of our knowledge, has not been previously studied in the context of CATE estimation. We will revise the abstract to better reflect our contribution.
>
> **Response to W2**
>
> Thank you for the insightful question. You are right that if the text used at inference time is affected by the treatment and/or outcome (e.g., discharge notes written after given treatment decisions), it may act as a post-treatment variable. Conditioning on such variables can induce post-treatment bias, as they may lie on causal paths from treatment to outcome (e.g., mediators or colliders).
>
> Hence, we agree that, in real-world datasets, not all observed covariates should be treated as confounders—some may indeed be **post-treatment** variables (e.g., mediators) and should not be used for adjustment. We will thus add a discussion with practical considerations to help practitioners identify post-treatment variables, which should not be used for adjustment.
>
> In our paper, we explicitly define $X$ as the set of **pre-treatment** variables that serve as confounders, in line with standard assumptions in the CATE estimation literature. Our method assumes that only such pre-treatment variables—whether structured or textual—are used for adjustment.
>
> In practice, it is crucial for practitioners to apply domain knowledge to identify and exclude post-treatment variables from the adjustment set. This is typically feasible in real-world clinical settings, as most records (including textual notes) are timestamped, making it possible to determine whether a variable was recorded before or after treatment assignment.
>
> **Response to W3**
>
> Thank you for the comment. While it may be intuitively expected that the naïve estimator is biased under inference-time text confounding, the goal of Lemma 4.1 is to formally formalize the pointwise bias.
>
> Lemma 4.1 provides an explicit expression that quantifies how the omission of confounding information (when using $T$ instead of $X$) affects the estimated treatment effect. This formal result is important for rigorously establishing the limitations of naïve estimators that use text representations without proper adjustment.
>
> We provide this intuition in Remark 4.2, which highlights that, in many real-world settings, the text variable $T$ often captures only a subset of the information in $X$, resulting in $Y(a) \not\perp A \mid T$. As a result, the naïve estimator is biased, and this bias is typically positive under common conditions. We will revise the paper to better emphasize that Lemma 4.1 is not just stating that bias exists, but rather providing a precise mathematical form of that bias.
>
> **Response to W4**
>
> Thank you for the insightful comment. We agree that identifying $\tau^x(x)$ at test time would be even more interesting.  However, in our setting, confounder $X$ is not accessible at inference time, which makes identification of $\tau^x(x)$ fundamentally impossible.
>
> We appreciate your suggestion and have incorporated it. We **conducted the comparison between estimated $\hat{\tau}^t(t)$ against the ground-truth $\tau^x(x)$** in in-sample data. We report both the averaged bias $\mathbb{E}\left[\hat{\tau}^t(t) - \tau^x(x)\right]$ and the root mean squared error (RMSE) $\sqrt{\mathbb{E}\left[\left(\hat{\tau}^t(t) - \tau^x(x)\right)^2\right]}$ as metrics.  The results for both the IST and MIMIC-III datasets across several baselines are below.
>
> | Method         | Averaged Bias | RMSE  |
> |----------------|----------------|--------|
> | TBE-Slearner   | 0.0628         | 0.198 |
> | TBE-Tlearner   | 0.0621         | 0.197 |
> | TBE-DRlearner  | 0.0495         | 0.186 |
> | Ours (TCA)     | **0.0140**     | **0.114** |
>
> | Method         | Averaged Bias | RMSE  |
> |----------------|----------------|--------|
> | TBE-Slearner   | 0.0763         | 0.283 |
> | TBE-Tlearner   | 0.0759         | 0.282 |
> | TBE-DRlearner  | 0.0671         | 0.265 |
> | Ours (TCA)     | **0.0204**     | **0.177** |
>
>
> This comparison provides empirical evidence that our method produces consistent and accurate approximations of $\tau^x(x)$, with minimal deviation, thereby validating the effectiveness of our approach.
>
> **Action**: We will add the above experiments to our revised paper.

---

> > ### Comment · Reviewer_kWL2 · 2025-08-03
> > **Thanks for the response**
> >
> > I thank the authors for the detailed response. I encourage authors to make the assumption that "all covariates are considered to be confounders" explicit in the revision. I do not have any further questions.

---

### Official Review · Reviewer_JPDz · 2025-06-26

**Clarity:** 4
**Significance:** 3
**Originality:** 3
**Rating:** 5
**Confidence:** 3

**Summary:**

Estimating treatment effects is an important task in many empirical sciences such as healthcare and political science. However, the information available when we seek to estimate treatment effects may not always be sufficient. Specifically, at inference time, we may only have access to text data that are noisy proxies of the actual confounders between treatment and outcome. The authors propose a novel methodology for estimating treatment effects when practitioners have access to only covariates at training time and only text data at inference time. The authors demonstrate the effectiveness of their method through semi-synthetic experiments.

**Questions:**

In Appendix D describing implementation details for the semi-synthetic experiments, the authors state that they used OpenAI’s APIs to generate surrogate text data. This implies that the authors passed covariate data from MIMIC-III through OpenAI’s API, which violates MIMIC-III’s terms and conditions (https://physionet.org/news/post/gpt-responsible-use). Can the authors verify whether they passed protected information through OpenAI’s API? I will raise my score if the authors are able to address this potential ethics violation.

What happens if the function $h$ for generating text data from covariates differs between training time and inference time? For instance, suppose the LLMs generate text data significantly differently from the way health practitioners write their notes. Would the authors expect downstream estimates for the treatment effect to be biased in this case?

What happens if, at inference time, text data is affected by the treatment and/or outcome? For instance, suppose a practitioner used discharge notes as $T$ at inference time, then the text data may likely be a child of both $A$ and $Y$, and conditioning on $T$ will induce confounding between the treatment and outcome. Is this something that practitioners need to worry about when considering the authors’ method? What are steps that practitioners can take to mitigate this concern?

In datasets such as MIMIC-III, clinical text data in addition to covariate data may be available at training time. In such a scenario, would the authors’ proposed method still apply by using the available text data directly to learn $f_\theta$? Would it still be necessary to generate surrogate text data $\tilde{T}$ in this case? Would the variability of downstream estimates for the conditional average treatment effect be improved in this case?

Does line 198 have a typo? Instead of “Y(1) if the treated”, did the authors mean to write “Y(1) if treated”?

Does line 214 have a typo? The final sentence seems to be incomplete and does not have any ending punctuation.

**Ethical Concerns:**

["Major Concern: Data privacy, copyright, and consent"]

**Final Justification:**

The authors take a principled approach to address a realistic problem that occurs when covariate data is available and text data is not available at training time but covariate data is not available and text data is available at testing time. The paper is well-written, and, through data studies, the authors show the effectiveness of their method. Although there was an initial ethics concern regarding the use of protected healthcare data in OpenAI's API, the authors have addressed this concern and verified that their data anlysis conforms with the terms and conditions of their dataset. Thus, I recommend acceptance of this paper.

**Limitations:**

Yes.

**Paper Formatting Concerns:**

None.

**Quality:**

3

**Strengths And Weaknesses:**

Strengths

One clear strength of this paper is its clarity. The authors explain clearly how their problem setting differs from problem settings considered previously in the literature. They also give a clear example (Figure 1) of when covariate data may be available at training time but not at inference time. Throughout the paper, the authors also do a great job introducing their problem setup, why naive approaches fail, and the assumptions required for their novel TCA framework to be valid.

The observation that generating synthetic text data at training time to learn a function on how average treatment effects vary based on text data to aid in the estimation of treatment effects at inference time is an interesting observation. As large language models improve in quality, it is great to see methods such as the authors’ that use them without relying on large language models’ reasoning abilities, which has been shown to be limited and flawed.

Previous work that considers text data as a proxy for estimating treatment effects requires two proxies and two separate and independent pieces of text data. In the authors’ method, they only require one instance of text data. Even though their method requires a training dataset with access to the ground truth covariates, a condition not required in previous work, the authors’ method applies to a new, and realistic, scenario and is thus valuable to the community. Practitioners now have more methods to choose from depending on the specific dataset that they have.

The semi-synthetic experiments are promising and demonstrate how different prompts to large language models affect the downstream accuracy of the authors’ proposed method pipeline. Table 2 also demonstrates that potential biases that may be present in LLMs do not significantly impact the fairness of downstream estimates for the treatment effect.

Weaknesses

I have no major complaints regarding this paper, but I have some questions below that I hope the authors can answer and discuss.

---

> ### Author Rebuttal · Authors · 2025-07-31
>
> Thank you for your positive review and your helpful comments! We are very happy to answer your questions and improve our paper as a result.
>
> ### Response to Questions
>
> **Answer to Q1**
>
> Thank you for your important question.
>
> We have access to ChatGPT Edu through our university, which is SOC 2 Type II compliant [1]. This gives us enhanced privacy controls not available in the standard API, free version, or even ChatGPT Team.
>
> As part of this setup, we have applied for a Business Associate Agreement (BAA) [2] to gain access to Zero Data Retention (ZDR). With ZDR enabled, the store parameter for /v1/responses and v1/chat/completions will always be treated as false [3]. In this setup, OpenAI does not retain any request or response data after processing—meaning there is no storage, no logging, and, more importantly, no human review.
>
> This configuration provides a privacy approach that is functionally equivalent to the one recommended by PhysioNet [4], where Azure OpenAI is used with human review explicitly disabled.
>
> => We confirm that our analysis is thus HIPAA-compliant and thus in line with the PhysioNet rules.
>
> [1] https://openai.com/index/introducing-chatgpt-edu/
>
> [2]https://help.openai.com/en/articles/8660679-how-can-i-get-a-business-associate-agreement-baa-with-openai-for-the-api-services
>
> [3] https://platform.openai.com/docs/guides/your-data#zero-data-retention
>
> [4] https://physionet.org/news/post/gpt-responsible-use
>
>
>
> **Answer to Q2**
>
> Thank you for raising this important point.
>
>  We acknowledge that LLM-generated text may not perfectly align with real-world text distributions and that a domain shift between training-time and inference-time text can occur in practice. Such shifts can affect methods’ performance.
>
> To evaluate this scenario, we conducted robustness experiments by varying the LLM prompt strategies used to generate text across datasets (Figure 3(b)). This introduces a controlled distribution shift in the text inputs. Despite these differences, our method consistently outperforms all baselines, demonstrating its robustness to moderate levels of text distribution mismatch.
>
> While discrepancies between the surrogate text $\tilde{T}$ and the true inference-time distribution $T$ may exist, they can potentially be mitigated through domain adaptation techniques. For instance, if a small sample of real-world text $T$ is available, one could adapt the generation process of $\tilde{T}$ to better align with it. We consider this a promising direction for future work.
>
> For example, there are several potential e strategies for extending our approach when a few text samples from the test distribution are available:  (i) In-context alignment: One strategy is to incorporate these additional samples directly into the prompt, guiding the LLM with instructions such as: "Here are example descriptions. Please formulate your descriptions in a similar style, level of detail, etc." (ii) Alignment via fine-tuning: Anothe strategy is fine-tuneing the LLM by introducing a loss term that minimizes the distributional distance between generated text and the test distribution samples—for example, using Wasserstein distance or Maximum Mean Discrepancy (MMD) as the alignment loss.
>
>
> **Answer to Q3**
>
> Thank you for highlighting this important point. We fully agree that if text data is generated after treatment assignment or outcome realization, it may act as a **post-treatment** variable (e.g., mediators), and adjusting for such variables can introduce collider bias or post-treatment bias.
>
> In our work, the $T$ used is only the set of **pre-treatment** variables that serve as confounders. In practice, it is crucial for practitioners to apply domain knowledge to identify and exclude post-treatment text from the adjustment set for inference data. This is typically feasible in real-world clinical settings, as most health records are timestamped, making it possible to determine whether a variable was recorded before or after treatment assignment.
>
> **Answer to Q4**
>
> Thank you for the insightful question. Yes, if clinical text data is available at training time, our method can still be applied.
>
>
> If the clinical text contains additional confounding information that is not captured in the structured covariates, we can include this text together with the covariates $X$ as an additional input to the text generation (i.e., the prompt could look like “A patient with features $X$ and the following reported self-reported symptoms[...]”) in the estimation procedure to avoid confounding bias. Further, this additional information helps to improve the accuracy of CATE estimation in downstream estimates. This flexibility resembles another strength of our method.
>
>
>  However, if the clinical text contains only confounding information that is also captured in $X$ or is less informative, but follows a different distribution compared to the inference-time text (e.g., different style or domain), then this text contains only noisy information that does not improve estimation. Thus, such text should not be considered, and we can apply our framework as is, using only the covariates $X$, which thus still requires generating surrogate text data.
>
> **Answer to Q5, 6**
> Thank you! We will fix the typos.

---

> > ### Author Response · Authors · 2025-08-05
> >
> > Thank you again for your effort in reviewing this paper and your constructive comments!
> > If the response addresses your concerns, we would greatly appreciate it if you would raise your score accordingly.
> > Thank you very much for reviewing our paper!

---

> > ### Comment · Reviewer_JPDz · 2025-08-05
> > **Response to Author Rebuttal**
> >
> > Thank you to the authors for their thoughtful response and interesting discussion. I would recommend the authors include their discussion on potential ways to extend their approach when some text samples from the training set are available in their conclusions section. Furthermore, the authors raise an interesting point that, when clinical text available at training time is not informative or follows a different distribution compared to inference-time text, such text data will not be informative and should thus be ignored. This point may be worth mentioning in the main text as well.
> >
> > As the authors have confirmed that their analysis is HIPAA-compliant and does not violate the terms and conditions of the MIMIC-III dataset, I will raise my score accordingly.

---

> > > ### Author Response · Authors · 2025-08-05
> > >
> > > Thank you so much for your response and for raising the score!
> > >
> > > We will include all suggested points in the paper. Thank you again for your effort in reviewing this paper and your constructive comments!

---

### Official Review · Reviewer_FZSB · 2025-07-09

**Clarity:** 4
**Significance:** 3
**Originality:** 3
**Rating:** 5
**Confidence:** 4

**Summary:**

The paper tackles the challenge (which they refer to as "inference time text confounding") when CATE estimation is done at inference time using text and confounders are not observed, but they are observed during training. The paper provides a framework for CATE estimation in this setting, and shows its real-world utility.

**Questions:**

(1) Can you please explain what are several real-world settings where inference-time text confounding applies, and specifically what are the interventions and outcomes we would care about that would motivate CATE estimation
(2) How does your method compare to strong baselines (e.g., CaML, and SIN [2])

[2] https://proceedings.neurips.cc/paper/2021/file/d02e9bdc27a894e882fa0c9055c99722-Supplemental.pdf

**Ethical Concerns:**

["NO or VERY MINOR ethics concerns only"]

**Final Justification:**

I thank the authors for their responses to my concerns, including w.r.t to novelty, baselines, as well as real world motivation. Authors have sufficiently addressed my concerns on baselines and novelty, and I appreciate the inclusion of new causal baselines to the paper. I have also thought through the author’s explanations on the real-world motivation of the inference time confounding problem. Upon reflection, I believe it is indeed a real-world problem since it is not always possible to quickly obtain covariates such as test results and diagnostics, and in emergency settings it is sometime needed to make decisions before obtaining covariates. However, I would strongly encourage authors in their final version to de-emphasize the telemedicine and chatbot motivation, since making personalized treatment decisions based on chatbots is extremely dangerous, as is telemedicine in which doctors are not soliciting information from patients but rather making decisions based on incomplete covariates. I have updated my score accordingly.

**Limitations:**

Yes.

**Quality:**

3

**Strengths And Weaknesses:**

Strengths
* The paper is very clearly written and the problem setting is clear and well-formulated
* Strong performance results compared to baselines. The robustness across varying conditions (confounder strengths and prompt strategies) enhances the credibility of results.
* The paper is highly reproducible

Weaknesses
* The TCA method seems to be a straightforward application of existing methodologies (meta-learners and DR pseudo-outcomes)
* While "Inference Time Text Confounding" is common in medical domains, it's not fully clear to me how this connects to CATE estimation. For example in Figure 1 I don't understand what CATE is. CATE estimation is useful when we are trying to decide on an intervention. But what is the *intervention*  in this case?
* Stronger baselines could be included. For example, the framework seems conceptually very similar to the CaML framework, in that you are (1) getting pseudo-outcomes as a pre-processing step, then (2) generating embeddings (3) Predicting the pseudo-outcomes from the embeddings. How does the performance of would this compare to using CaML out of the box?

[1]https://arxiv.org/pdf/2301.12292

---

> ### Author Rebuttal · Authors · 2025-07-31
>
> Thank you for your constructive review and your helpful comments! We are very happy to answer your questions and improve our paper as a result.
>
>
> ### Response to weakness
>
> **Response to W1**
>
> Thank you for the comment.  We would like to clarify that **TCA is not a straightforward application** of standard meta-learners or doubly robust (DR) learners.
>
> Our contributions go beyond existing methodologies in the following ways:
>
> (1) **Problem formalization:** We are the first to formally define the inference-time text confounding problem and show that naïve applications of existing meta-learners are biased. We derive **pointwise confounding bias** for these baselines, highlighting their limitations.
>
>
> (2) **Theoretical insight:** We provide the **identifiability result** connecting the target CATE under text ($\tau^t(t)$) to the CATE under structured confounders($\tau^x(x)$), and propose a two-stage estimation procedure to adjust for inference-time confounding. This theoretical connection is **non-trivial** and not addressed in prior work.
>
>
> (3) **Practical framework:** We introduce an **LLM-driven pipeline** to generate text and translate the theoretical result to a novel, practical framework for dealing with real-world conditions where structured data is unavailable at inference time.
>
>
> In contrast, **the baselines we include in our experiments are direct and straightforward applications of existing meta-learners using text as input**. Even though they share the same initial text-generation step (using LLMs), they **remain biased**, as demonstrated both **theoretically** (Lemma 4.1) and **empirically in our experiments** (Table 1). ] _A simple off-the-shelf combination of the text-generation step with existing DR-learners would be biased_.
>
> To show this empirically, **we performed new experiments against an off-the-shelf combination of our text-generation step and several more meta-learner baselines, including DR-learner**, X-learner, and R-learner. We call these baselines **“TBE-DR”**, “TBE-X”, “TBE-R”. Importantly, we made the experiment _fair_: we used the same setup for the LLMs, etc., so that the performance gains must be attributed to our framework. We again report PEHE scores on two datasets.
>
> | Dataset | TBE-DR | TBE-X | TBE-R | TCA (ours) |
> |---------|--------|-------|-------|------------|
> | IST     | 0.187  | 0.195 | 0.194 | 0.116      |
> | MIMIC   | 0.266  | 0.274 | 0.277 | 0.179      |
>
> => **Our TCA shows consistent improvements by a large margin**.
>
>
> Our method outperforms the TBE-DR baselines despite using the **same underlying estimation technique**—highlighting that the core contribution lies in **how we adjust for confounding**, not in the DR method itself.  Our method also outperforms other straightforward applications of existing meta-learner baselines. This demonstrates the effectiveness of our TCA framework and proves our method **differs substantially** from existing methodologies (meta-learners and DR pseudo-outcomes).
>
>
> **Action**: We will add the new experiments to our paper.
>
> **Response to W2**
>
> Thank you for the question.  In Figure 1, we illustrate a common setup from clinical datasets such as the IST dataset.
> The intervention in this example is a binary treatment: whether the patient receives the treatment assignment of aspirin allocation or not. The outcome is whether the patient experiences a recurrent stroke. Patient covariates, which are only observed directly during training time, include demographic variables (e.g., gender, age) and clinical measurements (e.g., presence of atrial fibrillation, blood pressure,  weakness in arms/legs). In this context, the CATE is the expected difference in the probability of having a stroke recurrence if a patient with covariates $X=x$ were given aspirin versus or not.
>
> => The motivation is that such CATE estimates help inform personalized treatment decisions: for which subgroups of patients is certain medicine beneficial, and for which it may not help much.
>
> The challenge we focus on in the paper is that, at inference time, when a new patient comes to ask about his/her case via telemedicine or remote healthcare consultations, or medical chatbots), the structured features may not be available (e.g., due to the limitation of diagnostic tests or clinical measurements,  sensor limitations, or privacy constraints), but this patient can describe his/her feeling. CATE predictions are made using only textual descriptions with self-reported symptoms of the patient.
>
> Our framework is designed for this setup: to address this **inference-time text confounding** setting by learning to estimate CATE from structured data at training time, and using text at inference time, while adjusting for confounding via the induced confounding information in the text.
>
>
> **Response to W3**
>
> Thank you for the suggestion. We would like to clarify that SIN [1] and CaML [2] focus on a **different setting** and that both are *not* meaningful baselines from a theoretical and empirical point.
>
> The baselines (SIN and CaML) focus on a **different causal graph**: these methods focus with _high-dimensional_ **treatments** (e.g., predicting treatment effects of a new drug by using the molecular structure or a text description), while our settings focus on _high-dimensional_ **confounders** (e.g., the text descriptions is what we use to adjust for confounding). Hence, the way of how high-dimensional information enters the machine pipeline is also different. This fundamental difference leads to **different causal graphs** and objectives.
>
> There are also some further differences: (i) Our method does not require additional information on complex treatments. (ii) We focus on mitigating inference-time confounding, whereas SIN and CaML are developed for estimating treatment effects of complex structured treatments (text or molecules). (iii) We additionally assume that text is only available at inference time, not at training time (while, for SIN and CaML, it is mandatory that text is available at training time).  Especially because of reason (iii), both baselines are **not directly applicable** to our setting.
>
> In addition to the theoretical explanation, we would like to provide empirical evidence. Although these methods are not directly applicable to our setting, we adapted their architectures for comparison. This means we use their architectures but integrate them into our LLM pipeline (e.g., we combine their architecture with the text generation step from our TCA framework). Further, for SIN[1], we use their method design to construct pseudo-outcomes and then regress on the text generated by LLMs to simulate missing treatment information. For CaML[2], as it is a zero-shot approach that requires additional structured information for a new treatment as input, we cannot apply it in its original form. Instead, we take its model architecture to predict CATE. The results for both IST and MIMIC-III datasets are below (reported: PEHE). **The results show that our method outperforms the baselines**.
>
>
> | Dataset         | CaML  | SIN   | TCA (ours) |
> |----------------|-------|-------|------------|
> | IST   | 0.201 | 0.195 | **0.116**      |
> | MIMIC  | 0.285 | 0.278 | **0.179**      |
>
>
>
> **Action**: We will add the above comparison to our revised paper.
>
> Reference:
>
> [1] Kaddour, J., Zhu, Y., Liu, Q., Kusner, M.J. and Silva, R. Causal effect inference for structured treatments. Advances in Neural Information Processing Systems, 2021.
>
> [2] Nilforoshan, H., Moor, M., Roohani, Y., Chen, Y., Šurina, A., Yasunaga, M., Oblak, S. and Leskovec, J.. Zero-shot causal learning. Advances in Neural Information Processing Systems, 2023.

---

> > ### Author Response · Authors · 2025-08-05
> >
> > Thank you again for your effort in reviewing this paper and your constructive comments! If the response addresses your concerns, we would greatly appreciate it if you would raise your score accordingly. Thank you very much for reviewing our paper!

---

> > ### Author Response · Authors · 2025-08-08
> >
> > We sincerely thank you for your time and effort in reviewing our paper. As the author–reviewer discussion period is nearing its end, we would be happy to clarify any remaining questions or concerns you may have.
> >
> > If our response has addressed your concerns, we would be grateful if you could update your score accordingly.
> > Thank you again for your thoughtful review.

---

### Note · Authors · 2025-08-13

Dear AC and reviewers,

Thank you for reviewing our paper. We are pleased that the overall feedback has been positive.

We have addressed reviewers’ concerns, and in the camera-ready version, we will:

1. Include the clarification related to the MIMIC III dataset.

2. Make explicit that $X$ denotes the set of pre-treatment variables that serve as confounders.

3. Add our new experiments from the rebuttal, which combined our text-generation pipeline with a standard DR-Learner (TBE-DR) and showed that our method outperformed TBE-DR by a large margin.

Our work differs substantially from prior approaches: we **formally define the inference-time text confounding problem**, derive the pointwise confounding bias of existing methods, and propose a novel framework for unbiased CATE estimation. The significance of our approach is supported by **strong** and state-of-the-art **empirical** performance.

Thank you again for your time and effort in reviewing our paper.

---

### Decision · Program_Chairs · 2025-09-17

**Decision:**

Accept (poster)

**Comment:**

All reviewers agree that this paper is clear, well-motivated and relevant in the context of the estimation of treatment effects. Two of the reviewers assigned a clearly positive score, particularly highlighting the novelty of the approach, the technical strength and the  real-world significance of causal effect estimation under test time text confounders.
The other two reviews are in the "borderline" area. These two reviewers mentioned some weaknesses, such as (some) weakly supported assumptions concerning data availability during training time and the (potentially) unrealistic assumption that all covariates are considered to be confounders. In the rebuttal, however the authors could successfully address most of these concerns in a plausible way. Therefore, I think that for this paper the positive aspects outweigh the weaknesses, and recommend acceptance of this paper